# Data-Efficient Molecular Generation with Hierarchical Textual Inversion

## Abstract

Developing an effective molecular generation framework even with a limited number of molecules is often important for its practical deployment, e.g., drug discovery, since acquiring task-related molecular data requires expensive and time-consuming experimental costs. To tackle this issue, we introduce *Hierarchical textual Inversion for Molecular generation* (HI-Mol), a novel data-efficient molecular generation method. HI-Mol is inspired by a recent textual inversion technique in the visual domain that achieves data-efficient generation via simple optimization of a new single text token of a pre-trained text-to-image generative model. However, we find that its naïve adoption fails for molecules due to their complicated and structured nature. Hence, we propose a hierarchical textual inversion scheme based on introducing low-level tokens that are selected differently per molecule in addition to the original single text token in textual inversion to learn the common concept among molecules. We then generate molecules using a pre-trained text-to-molecule model by interpolating the low-level tokens. Extensive experiments demonstrate the superiority of HI-Mol with notable data-efficiency. For instance, on QM9, HI-Mol outperforms the prior state-of-the-art method with $50\times$ less training data. We also show the efficacy of HI-Mol in various applications, including molecular optimization and low-shot molecular property prediction.

## 1 Introduction

Finding novel molecules has been a fundamental yet crucial problem in chemistry (Xue et al., 2019; Xu et al., 2019b) due to its strong relationship in achieving important applications, such as drug discovery (Segler et al., 2018; Bongini et al., 2021) and material design (Hamdia et al., 2019; Tagade et al., 2019). However, generating molecules poses a challenge due to their highly structured nature and the vast size of the input space (Drew et al., 2012). To tackle this issue, several works have considered training deep generative models to learn the molecule distribution using large molecular datasets (Ahn et al., 2022; Jo et al., 2022). This is inspired by the recent breakthroughs of generative models in other domains, e.g., images and videos (Rombach et al., 2022; Singer et al., 2022; Yu et al., 2023), in learning high-dimensional and complex data distribution. Intriguingly, such deep molecular generation methods have demonstrated reasonable performance (Jin et al., 2018; 2020; Ahn et al., 2022) on the large-scale benchmarks (Ramakrishnan et al., 2014; Polykovskiy et al., 2020a) in finding chemically valid and novel molecules, showing great potential to solve the challenge.

Unfortunately, existing molecular generation frameworks tend to fail in limited data regimes (Guo et al., 2022). This restricts the deployment of existing approaches to practical scenarios, because task-related molecular data for the target real-world applications are often insufficient to train such molecular generative models. For example, drug-like molecules for a specific organ are inherently scarce in nature (Schneider & Fechner, 2005; Altae-Tran et al., 2017), and the drug-likeness of each candidate molecule should be verified through years of extensive wet experiments and clinical trials (Drews, 2000; Hughes et al., 2011). This time-consuming and labor-intensive data acquisition process of new task-related molecules (Stanley et al., 2021) limits the number of available training data for a model to learn the desired molecule distribution. Thus, it is often crucial to develop a *data-efficient molecular generation* framework, yet this direction has been overlooked in the field of deep molecular generation (Guo et al., 2022) despite its importance in achieving practical applications.

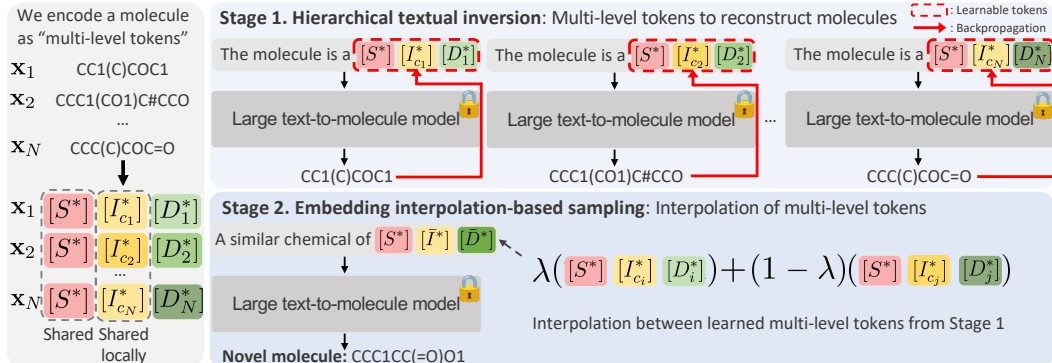

Figure 1: Overview of our HI-Mol framework. (1) Hierarchical textual inversion: we encode the features of molecules into multi-level token embeddings. (2) Embedding interpolation-based sampling: we generate novel molecules using interpolation of low-level token embeddings.

Meanwhile, recent works in text-to-image generation have explored the problem of low-shot (or personalized) generation using the power of large pre-trained models trained on a massive amount of data (Ruiz et al., 2022; Wei et al., 2023). In particular, Gal et al. (2022) propose a *textual inversion* using pre-trained text-to-image diffusion models—given a small set of images, they show that the common concepts among them can be learned effectively by optimizing a single text token under the frozen diffusion model, where the learned token can be used for the desired generation.

Considering the recent success of large-scale pre-trained text-to-molecule models (Edwards et al., 2022), what we ask in this paper is: *can textual inversion be exploited to enable data-efficient molecular generation with large-scale pre-trained text-to-molecule models?* However, we find that naïve adoption of textual inversion fails to achieve the goal, due to the highly complicated and structured nature of molecules (see Figure 2). To exploit textual inversion for data-efficient molecular generation, we suggest considering the unique aspects of the molecule carefully in its adoption.

**Contribution.** We introduce a novel data-efficient molecular generation method, coined **H**ierarchical textual **I**nversion for **Mol**ecular generation (**HI-Mol**). Specifically, HI-Mol is composed of two components (see Figure 1 for the overall illustration):

- *Hierarchical textual inversion:* We propose a molecule-specialized textual inversion scheme to capture the hierarchical information of molecules (Alexander et al., 2011). In contrast to textual inversion for the visual domain that optimizes a single shared token on given data, we design multi-level tokens for the inversion so that some of the low-level tokens are selected differently per molecule. Thus, the shared token learns the common concept among molecules and low-level tokens learn molecule-specific features. This low-level token selection does not require any specific knowledge of each molecule and can be achieved completely in an unsupervised manner.

- *Embedding interpolation-based sampling:* We present a molecule sampling scheme that utilizes the multi-level tokens optimized in the inversion stage. Our main idea is to use low-level tokens in addition to the shared token for molecular generation. In particular, we consider using the interpolation of two different low-level token embeddings for generation. The mixing approach is designed to extensively utilize the information of given molecules, and thus effectively alleviates the issue of the limited number of available molecules that lie in the target distribution.

We extensively evaluate HI-Mol by designing several data-efficient molecular generation tasks on the datasets in the MoleculeNet benchmark (Wu et al., 2018) and on the QM9 dataset (Ramakrishnan et al., 2014). For instance, in the HIV dataset in MoleculeNet, HI-Mol improves Frechet ChemNet Distance (Preuer et al., 2018, FCD) and Neighborhood Subgraph Pairwise Distance Kernel MMD (Costa & De Grave, 2010, NSPDK) as $20.2 \rightarrow 16.6$, and $0.033 \rightarrow 0.019$ (respectively, lower values are better) from prior arts. HI-Mol also achieves much better active ratio (higher is better) by improving the previous state-of-the-art as $3.7 \rightarrow 11.4$. We also show the strong data-efficiency of HI-Mol. For instance, on QM9, HI-Mol already outperforms the previous state-of-the-arts, e.g., STGG (Ahn et al., 2022) by $0.585 \rightarrow 0.434$ in FCD, with $50\times$ less training data. Finally, we validate the effectiveness of HI-Mol on several downstream tasks including the molecular optimization for PLogP on the ZINC dataset (Irwin et al., 2012) and the low-shot molecular property prediction on MoleculeNet.

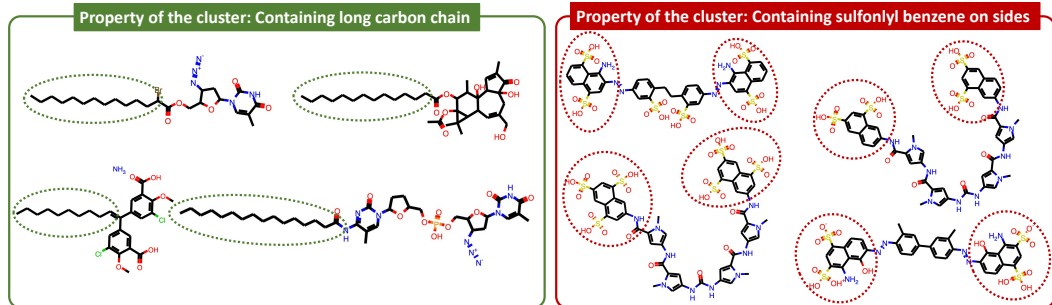

Figure 2: Visualizations of molecules in two clusters obtained from the unsupervised clustering in Eq. (1) on the HIV dataset (Wu et al., 2018). We note that the molecules often have very different structures, e.g., long carbon chains (left) and sulfonyl benzene groups (right), and thus the naïve application of textual inversion with a single shared token does not perform well (see Table 6).

## 2 RELATED WORK

**Molecular generation.** Most molecular generation methods fall into three categories based on different representations of molecules. First, there exist many attempts (Shi et al., 2020; Zang & Wang, 2020; Niu et al., 2020; Luo et al., 2021; Liu et al., 2021; Jo et al., 2022; Luo et al., 2022; Guo et al., 2022; Hoogeboom et al., 2022; Zhang et al., 2023; Vignac et al., 2023) to formalize molecular generation as a graph generation problem by representing each molecule as an attributed graph. Next, there are several fragment-based methods (Jin et al., 2018; Kong et al., 2022; Geng et al., 2023), which define a dictionary of fragments, e.g., functional groups. Each molecule is represented as a tree structure of dictionary elements and the distribution of connected fragments is then modeled. Finally, there are approaches (Gómez-Bombarelli et al., 2016; Liu et al., 2018; Flam-Shepherd et al., 2022; Ahn et al., 2022) that utilize the Simplified Molecular-Input Line-Entry System (SMILES, Weininger, 1988) representation to write molecules as strings and learn the distribution in this string space.

**Molecular language model.** Following the recent progress in large language models (Raffel et al., 2020; Brown et al., 2020; Touvron et al., 2023), there exist several attempts to train molecular language models (Fabian et al., 2020; Bagal et al., 2021; Christofidellis et al., 2023). Specifically, these works exploit popular language model architectures to have pre-trained models for molecules, based on the SMILES (Weininger, 1988) representation $\mathrm{SMILES}(\mathbf{x})$ that interprets a given molecule $\mathbf{x}$ as a string. In particular, MolT5 (Edwards et al., 2022) proposes to fine-tune a large text-to-text language model, T5 (Raffel et al., 2020), with SMILES representations of large-scale molecular data and text description-SMILES pair data to have a text-to-molecule model. Notably, it results in a highly effective pre-trained model for molecules, demonstrating superior performance across text-to-molecule generation tasks. Inspired by its success, we use the Large-Caption2Smiles model trained with this MolT5 approach for our goal of data-efficient molecular generation.

**Low-shot generation.** There have been substantial efforts in the generative model literature to design a low-shot generation framework for generating new samples from a given small number of data. Intriguingly, recent works on large-scale text-to-image diffusion models have surprisingly resolved this challenge, even enabling "personalization" of the model at a few in-the-wild images through simple optimization schemes that update only a few parameters (Gal et al., 2022; Cohen et al., 2022; Wei et al., 2023). In particular, textual inversion (Gal et al., 2022) exhibits that the personalization of large-scale text-to-image diffusion models can be achieved even with a very simple optimization of a single additional text token without updating any pre-trained model parameters.

In contrast to the recent advances of low-shot generation in the image domain, developing a low-shot (or data-efficient) molecular generation framework is relatively under-explored despite its practical importance (Altae-Tran et al., 2017; Guo et al., 2022). Hence, our method tackles this problem by designing a molecule-specific textual inversion method using the recent large-scale text-to-molecule models. Specifically, due to our unique motivation to consider "hierarchy" of molecular structures (Alexander et al., 2011), our method effectively learns the molecule distribution of low-shot molecules with diverse molecular structures, while the applications of prior works, e.g., Guo et al. (2022), are limited to structurally similar low-shot molecules such as monomers and chain-extenders.

# 3 HI-MOL: HIERARCHICAL TEXTUAL INVERSION FOR MOLECULAR GENERATION

In Section 3.1, we provide an overview of our problem and the main idea. In Section 3.2, we provide descriptions of textual inversion to explain our method. In Section 3.3, we provide a component-by-component description of our method.

## 3.1 PROBLEM DESCRIPTION AND OVERVIEW

We formulate our problem of *data-efficient molecular generation* as follows. Consider a given molecular dataset $\mathcal{M} := \{\mathbf{x}_n\}_{n=1}^N$, where each molecule $\mathbf{x}_n$ is drawn from an unknown task-related molecule distribution $p(\mathbf{x}|\mathbf{c})$. Here, $\mathbf{c}$ represents the common underlying chemical concept among molecules in the dataset for the target task, e.g., blood-brain barrier permeability or ability to inhibit HIV replication. We aim to learn a model distribution $p_{\mathrm{model}}(\mathbf{x})$ that matches $p(\mathbf{x}|\mathbf{c})$, where the number of molecules $N$ in the dataset is small, e.g., $N = 691$ in the BACE dataset.

To solve this problem, we take the recent approach of textual inversion (Gal et al., 2022) from the text-to-image diffusion model literature—a simple yet powerful technique in low-shot image generation that learns a common concept in given images as a single token in text embedding space. Similarly, we aim to learn the common concept of molecules as text tokens and use them for our target of data-efficient generation. However, exploiting this approach for our goal faces several challenges, mainly due to the unique characteristics of molecules differentiated from images. First, it is yet overlooked *which* of the large-scale model for molecules is beneficial to achieve textual inversion for molecules, like the success of text-to-image diffusion models in achieving successful inversion in the image domain. Moreover, molecules have a very different structural nature from images—unlike images, molecules with similar semantics often have entirely different structures (see Figure 2), making it difficult to simply learn the common concept as a single text token. Our contribution lies in resolving these challenges by adopting molecule-specific priors into the framework to enjoy the power of textual inversion techniques in achieving data-efficient molecular generation.

## 3.2 PRELIMINARY: TEXTUAL INVERSION

Recent text-to-image generation methods have proposed textual inversion (Gal et al., 2022), which aims to learn a common concept $\mathbf{c}$, i.e., the distribution $p(\mathbf{x}|\mathbf{c})$, from a small set of images and use it for the concept-embedded (or personalized) generation. To achieve this, they optimize a *single* text embedding of a token $[S^*]$ shared among images to learn $\mathbf{c}$ using a pre-trained frozen text-to-image diffusion model $f_{\mathtt{t2i}}$. Specifically, they put $[S^*]$ with a short text description, e.g., "A photo of $[S^*]$", as the text prompt to $f_{\mathtt{t2i}}$, and then optimize this token embedding using given images with the exact same training objective that is used for training $f_{\mathtt{t2i}}$. We propose to adapt the textual inversion framework into the data-efficient molecular generation framework based on the recent state-of-the-art large-scale pre-trained text-to-molecule generative model, MolT5 (Edwards et al., 2022).[1]

## 3.3 DETAILED DESCRIPTION OF HI-MOL

**Hierarchical textual inversion.** We first propose a molecule-specific textual inversion to learn the desired molecule distribution. Unlike prior textual inversion that assumes a single shared token $[S^*]$ only, we propose to use "hierarchical" tokens $[S^*], \{[I_k^*]\}_{k=1}^K, \{[D_n^*]\}_{n=1}^N$ (with parametrizations $\theta := (\mathbf{s}, \{\mathbf{i}_k\}_{k=1}^K, \{\mathbf{d}_n\}_{n=1}^N)$) by introducing additional intermediate tokens $\{[I_k^*]\}_{k=1}^K$ and detail tokens $\{[D_n^*]\}_{n=1}^N$ (with $K < N$). Such intermediate and detail tokens learn cluster-wise (high-level) and molecule-wise (low-level) features of the molecular dataset, respectively.

To learn these hierarchical tokens, we consider a frozen text-to-molecule model $f$, e.g., Large-Caption2Smiles (Edwards et al., 2022), to apply our proposed hierarchical textual inversion objective.

---

[1]We use SMILES strings as the representation of molecules because our method is built upon the state-of-the-art text-to-molecule model that utilizes SMILES strings, i.e., MolT5 (Edwards et al., 2022). However, our method is agnostic to the underlying molecule representation of the text-to-molecule models.

Specifically, we optimize $\theta$ by minimizing the following objective on the given molecular dataset $\mathcal{M}$:

$$\mathcal{L}(\theta; \mathbf{x}_n) := \min_{k \in [1, K]} \mathcal{L}_{\text{CE}}\Big(\texttt{softmax}\big(f(\text{``The molecule is a } [S^*][I_k^*][D_n^*]\text{''})\big), \texttt{SMILES}(\mathbf{x}_n)\Big), \quad (1)$$

where $\mathcal{L}_{\text{CE}}$ denotes cross-entropy loss and $\texttt{SMILES}(\mathbf{x}_n)$ is a SMILES (Weininger, 1988) string of $\mathbf{x}_n$. Thus, each $\mathbf{x}_n$ is interpreted as three tokens $[S^*][I_{c_n}^*][D_n^*]$, where we assign intermediate token index $c_n \in [1, K]$ (for given $\mathbf{x}_n$ and the corresponding $[D_n^*]$) during optimization to minimize the training objective $\mathcal{L}$ (see Eq. (1)). We note that the selection of $[I_{c_n}^*]$ is achieved in an unsupervised manner so that it does not require any specific information about each molecule. Intriguingly, we find that $[I_{c_n}^*]$ can learn some of the informative cluster-wise features through this simple selection scheme although we have not injected any prior knowledge of given molecular data (see Figure 2 for an example).

Our "multi-level" token design is particularly important for the successful inversion with molecules because molecules have a different nature from images that are typically used in the existing textual inversion method. Image inputs in the conventional textual inversion are visually similar, e.g., pictures of the same dog with various poses, whereas molecules often have entirely different structures even if they share the common concept, e.g., ability on the blood-brain membrane permeability (Wu et al., 2018). This difference makes it difficult to learn the common concept as a simple single token; we mitigate it by adopting hierarchy in the inversion scheme by incorporating the principle of chemistry literature highlighting that molecular data can be clustered hierarchically (Alexander et al., 2011).

**Embedding interpolation-based sampling.** We propose a sampling strategy from the learned distribution via hierarchical textual inversion. We find that sampling schemes used in existing textual inversion for images, e.g., putting a text prompt including $[S^*]$ such as "A similar chemical of $[S^*]$" into the molecular language model $f$, does not work well in molecular generation (see Table 6).

To alleviate this issue, we propose to utilize the learned intermediate tokens $\{[I_k^*]\}_{k=1}^K$ and detail tokens $\{[D_n^*]\}_{n=1}^N$ to sample from our target distribution. We consider the interpolation of each of intermediate tokens and detail tokens in the sampling process, i.e., we incorporate the hierarchy information of molecules which is obtained in our textual inversion. Specifically, we sample a novel molecule with random molecule indices $i, j$ sampled uniformly from $[1, \ldots, N]$ and a coefficient $\lambda$ drawn from a pre-defined prior distribution $p(\lambda)$ (see Appendix A for our choice of $p(\lambda)$):

$$(\bar{\mathbf{i}}, \bar{\mathbf{d}}) := \lambda(\mathbf{i}_{c_i}, \mathbf{d}_i) + (1 - \lambda)(\mathbf{i}_{c_j}, \mathbf{d}_j), \quad (2)$$
$$\mathbf{x} := f(\text{``A similar chemical of } [S^*][\bar{I}^*][\bar{D}^*]\text{''}),$$

where $[\bar{I}^*], [\bar{D}^*]$ indicate that we pass interpolated embeddings $\bar{\mathbf{i}}, \bar{\mathbf{d}}$ to $f$, respectively, and $c_n \in [1, K]$ is an index of the intermediate token of a given molecule $\mathbf{x}_n$, i.e., an intermediate token index that minimizes the training objective in Eq. (1).[2] This additional consideration of low-level tokens $\{[I_k^*]\}_{k=1}^K, \{[D_n^*]\}_{n=1}^N$ (as well as $[S^*]$) encourages the sampling process to exploit the knowledge from given molecular dataset extensively, mitigating the issue of scarcity of target molecules that lie in our desired molecule distribution and thus enables to generate high-quality molecules. We provide qualitative analysis on our embedding interpolation-based sampling scheme in Appendix I.

## 4 EXPERIMENTS

We extensively verify the superiority of HI-Mol by considering various data-efficient molecular generation scenarios. In Section 4.1, we explain our experimental setup. In Section 4.2, we present our main molecular generation results on MoleculeNet and QM9. In Section 4.3, we present results on downstream tasks, i.e., optimization and low-shot property prediction. Finally, in Section 4.4, we conduct some analysis and an ablation study to validate the effect of components of our method. We provide further ablation study and additional experimental results in Appendix E and F, respectively.

### 4.1 EXPERIMENTAL SETUP

**Datasets.** Due to the lack of benchmarks designed particularly for data-efficient molecular generation, we propose to use the following datasets for evaluating molecular generation methods under our problem setup. First, we consider three datasets in the MoleculeNet (Wu et al., 2018) benchmark

---

[2] We simply set the number of clusters $K$ as 10 in our experiments. Please see Appendix E for analysis on $K$.

Table 1: Quantitative results of the generated molecules on the three datasets (HIV, BBBP, BACE) in the MoleculeNet benchmark (Wu et al., 2018). We mark in Grammar if the method explicitly exploits the grammar of molecular data and thus yields a high Valid. score. The Active. score is averaged over three independently pre-trained classifiers. We compute and report the results using the 500 non-overlapping generated molecules to the training dataset. We set the highest score in bold. ↑ and ↓ indicate higher and lower values are better (respectively) for each metric.

| Dataset | Method | Class | Grammar | Active. ↑ | FCD ↓ | NSPDK ↓ | Valid. ↑ | Unique. ↑ | Novelty ↑ |
|---|---|---|---|---|---|---|---|---|---|
| HIV | GDSS (Jo et al., 2022) | Graph | ✗ | 0.0 | 34.1 | 0.080 | 69.4 | **100** | **100** |
| | DiGress (Vignac et al., 2023) | Graph | ✗ | 0.0 | 26.2 | 0.067 | 17.8 | **100** | **100** |
| | JT-VAE (Jin et al., 2018) | Fragment | ✓ | 0.0 | 38.8 | 0.221 | **100** | 25.4 | **100** |
| | PS-VAE (Kong et al., 2022) | Fragment | ✓ | 3.7 | 21.8 | 0.053 | **100** | 91.4 | **100** |
| | MiCaM (Geng et al., 2023) | Fragment | ✓ | 3.4 | 20.4 | 0.037 | **100** | 81.6 | **100** |
| | CRNN (Segler et al., 2018) | SMILES | ✗ | 3.3 | 29.7 | 0.064 | 30.0 | **100** | **100** |
| | STGG (Ahn et al., 2022) | SMILES | ✓ | 1.6 | 20.2 | 0.033 | **100** | 95.8 | **100** |
| | **HI-Mol (Ours)** | SMILES | ✗ | **11.4** | 19.0 | **0.019** | 60.6 | 94.1 | **100** |
| | **HI-Mol (Ours)** | SMILES | ✓ | **11.4** | 16.6 | **0.019** | **100** | 95.6 | **100** |
| BBBP | GDSS (Jo et al., 2022) | Graph | ✗ | 0.0 | 35.7 | 0.065 | 88.4 | 99.2 | **100** |
| | DiGress (Vignac et al., 2023) | Graph | ✗ | 8.2 | 17.4 | 0.033 | 43.8 | 94.6 | **100** |
| | JT-VAE (Jin et al., 2018) | Fragment | ✓ | 80.6 | 37.4 | 0.202 | **100** | 10.8 | **100** |
| | PS-VAE (Kong et al., 2022) | Fragment | ✓ | 84.9 | 17.3 | 0.039 | **100** | 91.6 | **100** |
| | MiCaM (Geng et al., 2023) | Fragment | ✓ | 82.0 | 14.3 | 0.021 | **100** | 89.4 | **100** |
| | CRNN (Segler et al., 2018) | SMILES | ✗ | 88.8 | 20.2 | 0.026 | 54.0 | **100** | **100** |
| | STGG (Ahn et al., 2022) | SMILES | ✓ | 89.1 | 14.4 | 0.019 | 99.8 | 95.8 | **100** |
| | **HI-Mol (Ours)** | SMILES | ✗ | 94.4 | 11.2 | 0.011 | 78.8 | 92.9 | **100** |
| | **HI-Mol (Ours)** | SMILES | ✓ | **94.6** | **10.7** | **0.009** | **100** | 94.2 | **100** |
| BACE | GDSS (Jo et al., 2022) | Graph | ✗ | 9.1 | 66.0 | 0.205 | 73.4 | **100** | **100** |
| | DiGress (Vignac et al., 2023) | Graph | ✗ | 21.1 | 26.7 | 0.102 | 16.4 | **100** | **100** |
| | JT-VAE (Jin et al., 2018) | Fragment | ✓ | 40.4 | 49.1 | 0.304 | **100** | 13.0 | **100** |
| | PS-VAE (Kong et al., 2022) | Fragment | ✓ | 57.3 | 30.2 | 0.111 | **100** | 75.6 | **100** |
| | MiCaM (Geng et al., 2023) | Fragment | ✓ | 56.2 | 18.5 | 0.060 | **100** | 64.2 | **100** |
| | CRNN (Segler et al., 2018) | SMILES | ✗ | 79.0 | 21.7 | 0.066 | 38.0 | **100** | **100** |
| | STGG (Ahn et al., 2022) | SMILES | ✓ | 42.9 | 17.6 | 0.053 | **100** | 94.8 | **100** |
| | **HI-Mol (Ours)** | SMILES | ✗ | **81.0** | 16.4 | 0.052 | 71.0 | 69.9 | **100** |
| | **HI-Mol (Ours)** | SMILES | ✓ | 80.4 | **14.0** | **0.039** | **100** | 74.4 | **100** |

(originally designed for activity detection): HIV, BBBP, and BACE, which have a significantly smaller number of molecules than popular molecular generation benchmarks (Sterling & Irwin, 2015; Polykovskiy et al., 2020b). For example, BACE includes only 691 active molecules. With only the active molecules in each dataset, we construct tasks to generate novel molecules that share the same chemical concept, e.g., blood-brain membrane permeability for the BBBP dataset.

Moreover, we also utilize the QM9 dataset (Ramakrishnan et al., 2014) for our experiments to show the data-efficiency of HI-Mol. Specifically, we train our method with an extremely small subset of the entire QM9 training split, e.g., 2%, where other baseline methods are trained with the whole training split (105k molecules). We provide more details about the datasets in Appendix B.

**Evaluation setup.** To evaluate the quality of the generated molecules, we consider six metrics that represent diverse aspects which are critical to the evaluation of the generated molecules, e.g., similarity to the target distribution, uniqueness, and novelty. We incorporate some well-known metrics, such as those used in Jo et al. (2022), as well as introducing a new metric "Active ratio":

- **Active ratio**[3] (**Active.**): Our proposed metric, measuring the ratio of the valid generated molecules that are active, i.e., satisfying the target property for the relevant task.

- **Fréchet ChemNet Distance** (**FCD**, Preuer et al., 2018): Metric for measuring the distance between the source distribution and the target distribution using pre-trained ChemNet.

- **Neighborhood Subgraph Pairwise Distance Kernel MMD** (**NSPDK**, Costa & De Grave, 2010): Another metric for measuring the gap between source and the target distributions, based on algorithmic computation using graph-based representations of molecules.

- **Validity (Valid.):** The ratio of the generated molecules that have the chemically valid structure.

---

[3]For reliable evaluation with our metric, we avoid the overlap between the generated molecules and the training data used for generation methods by ignoring the molecule if it is contained in this dataset. Hence, the Novelty score is 100 for all MoleculeNet experiments since all samples are different from the training set (see Table 1 for an example). We provide the detailed description of this metric in Appendix C.

Table 2: Qualitative results of the generated molecules on the two datasets (HIV, BBBP) of the MoleculeNet benchmark (Wu et al., 2018). We visualize the generated molecules from each method that has the maximum Tanimoto similarity with a given anchor molecule. We report the similarity below each visualization of the generated molecule. We set the highest similarity in bold.

| Dataset | DiGress (Vignac et al., 2023) | MiCaM (Geng et al., 2023) | STGG (Ahn et al., 2022) | HI-Mol (Ours) | Train |
|---------|---------|---------|---------|---------|-------|
| HIV | 0.154 | 0.146 | 0.157 | **0.326** | |
| BBBP | 0.238 | 0.247 | 0.246 | **0.505** | |

- **Uniqueness (Unique.)**: Diversity of the generated molecules based on the ratio of different samples over total valid molecules earned from the generative model.

- **Novelty**: Fraction of the valid molecules that are not included in the training set.

**Baselines.** We mainly consider the following methods for evaluation: GDSS (Jo et al., 2022), DiGress (Vignac et al., 2023), DEG (Guo et al., 2022), JT-VAE (Jin et al., 2018), PS-VAE (Kong et al., 2022), MiCaM (Geng et al., 2023), CRNN (Segler et al., 2018), and STGG (Ahn et al., 2022). For evaluation on QM9, we also consider GraphAF (Shi et al., 2020), GraphDF (Luo et al., 2021), MoFlow (Zang & Wang, 2020), EDP-GNN (Niu et al., 2020), and GraphEBM (Liu et al., 2021), following the recent works (Jo et al., 2022; Luo et al., 2022). We provide more details of the baselines in Appendix D.

## 4.2 MAIN RESULTS

**Generation on MoleculeNet.** Table 1 summarizes the quantitative results of the generated molecules on the HIV, BBBP, and BACE datasets in the MoleculeNet benchmark (Wu et al., 2018). Our method consistently outperforms other generation methods in terms of Active ratio, FCD, and NSPDK scores on all three datasets. We note that the improvements in these scores are particularly crucial for the deployment of the molecular generation method. For example, the superior Active ratio of HI-Mol, e.g., $3.7 \rightarrow 11.4$ on the HIV dataset, indicates that the generated molecules are more likely to exhibit the desired activeness. Our method also significantly improves the FCD metric on the HIV dataset from $20.2 \rightarrow 19.0$, and this indicates the effectiveness of HI-Mol in generating more faithful molecules that lie in the target distribution. We provide qualitative results in Table 2 by visualizing some of the generated molecules from each dataset. One can observe that the generated molecules by HI-Mol capture several crucial common substructures, e.g., many ester groups, while introducing the novel components, e.g., 4-membered ring, due to our interpolation-based sampling scheme.

We also propose a simple algorithm to modify the generated invalid SMILES by correcting invalid patterns[4] without a computational overhead. By applying this algorithm, we convert all invalid SMILES to valid ones, therefore, Validity becomes 100. In particular, the modified molecules further improve the overall metrics, e.g., FCD by $19.0 \rightarrow 16.6$ and $11.2 \rightarrow 10.7$ in the HIV and BBBP dataset, respectively. This indicates the modified SMILES indeed represent molecules from the desired distribution and further highlights the superior quality of our generated molecules.

**Generation on QM9.** In Table 3, we report the quantitative results of the generated molecules from each method. Here, we train our method with a limited portion of data, e.g., 2% and 10%, and then compare the results with the baselines that are trained with the entire dataset. Our model shows strong data-efficiency: only with a 2% subset of the training data, our method already outperforms the state-of-the-art baseline, STGG (Ahn et al., 2022), by $0.585 \rightarrow 0.430$ in FCD. Utilizing a 10% subset further improves the performance of HI-Mol, reducing the FCD by $0.430 \rightarrow 0.398$. In particular, compared with STGG, HI-Mol not only improves the FCD score but also shows a better Novelty score, which validates the capability of HI-Mol to find novel molecules from the target distribution.

---

[4]For example, we modify the invalid SMILES caused by the unclosed ring, e.g., C1CCC $\rightarrow$ CCCCC. Please see Appendix H for detailed algorithm. We mark in Grammar column when modification is applied for evaluation.

Table 3: Quantitative results of the generated molecules on the QM9 dataset (Ramakrishnan et al., 2014). We mark in Grammar if the method explicitly exploits the grammar of molecular data and thus yields a high Valid. score. Following the setup of Jo et al. (2022), we report the results using 10,000 sampled molecules. We denote the scores drawn from Luo et al. (2022) and Ahn et al. (2022) with (*) and (†), respectively. We mark (-) when the score is not available in the literature. We set the highest score in bold. ↑ and ↓ indicate higher and lower values are better (respectively) for each metric. For our method, we report the ratio of the number of samples of the dataset used for training.

| Method | Class | Grammar | FCD ↓ | NSPDK ↓ | Valid. ↑ | Unique. ↑ | Novelty ↑ |
|---|---|---|---|---|---|---|---|
| CG-VAE[†] (Liu et al., 2018) | Graph | ✓ | 1.852 | - | **100** | 98.6 | 94.3 |
| GraphAF (Shi et al., 2020) | Graph | ✗ | 5.268 | 0.020 | 67 | 94.5 | 88.8 |
| MoFlow (Zang & Wang, 2020) | Graph | ✗ | 4.467 | 0.017 | 91.4 | 98.7 | 94.7 |
| EDP-GNN (Niu et al., 2020) | Graph | ✗ | 2.680 | 0.005 | 47.5 | **99.3** | 86.6 |
| GraphDF (Luo et al., 2021) | Graph | ✗ | 10.82 | 0.063 | 82.7 | 97.6 | **98.1** |
| GraphEBM (Liu et al., 2021) | Graph | ✗ | 6.143 | 0.030 | 8.22 | 97.8 | 97.0 |
| GDSS (Jo et al., 2022) | Graph | ✗ | 2.900 | 0.003 | 95.7 | 98.5 | 86.3 |
| GSDM* (Luo et al., 2022) | Graph | ✗ | 2.650 | 0.003 | 99.9 | - | - |
| STGG[†] (Ahn et al., 2022) | SMILES | ✓ | 0.585 | - | **100** | 95.6 | 69.8 |
| **HI-Mol (Ours; 2%)** | SMILES | ✗ | 0.434 | **0.001** | 90.7 | 75.8 | **73.5** |
| **HI-Mol (Ours; 2%)** | SMILES | ✓ | 0.430 | **0.001** | **100** | 76.1 | 75.6 |
| **HI-Mol (Ours; 10%)** | SMILES | ✗ | 0.400 | 0.002 | 87.6 | 87.6 | 71.2 |
| **HI-Mol (Ours; 10%)** | SMILES | ✓ | **0.398** | **0.001** | **100** | 88.3 | 73.2 |

For an extensive comparison with the baselines which show high Novelty scores, e.g., GDSS (Jo et al., 2022), we adjust the sampling strategy slightly; we utilize a simple resampling strategy (which takes only 1.8 sec per molecule) and make the Validity, Uniqueness, and Novelty scores to 100 for a fair comparison in FCD with those methods. Even in this case, HI-Mol achieves FCD of 0.601, which outperforms all those baselines. We provide detailed results and discussions in Appendix G.

## 4.3 APPLICATIONS OF HI-MOL

**Molecular optimization.** We demonstrate the effectiveness of HI-Mol in molecular optimization, mainly following the experimental setup of Ahn et al. (2022). We train a conditional molecular generative model $p_{\mathrm{model}}(\mathbf{x}|\gamma)$ under the HI-Mol framework where $\gamma$ denotes the penalized octanol-water partition coefficient (PLogP). Then, we sample with a high $\gamma$ to generate molecules with high PLogP. In Table 4, our HI-Mol generates molecules with considerably high PLogP even when trained with only 1% of the entire

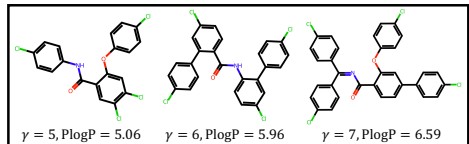

$\gamma = 5, \mathrm{PlogP} = 5.06$   $\gamma = 6, \mathrm{PlogP} = 5.96$   $\gamma = 7, \mathrm{PlogP} = 6.59$

Figure 3: Visualization of the generated molecules with condition $\gamma$. The maximum PLogP among the training molecules is 4.52.

training dataset. Here, we remark that solely maximizing the molecular property (such as PLogP) may generate unrealistic molecules (Ahn et al., 2022), e.g., unstable or hard-to-synthesize (see Appendix K). To address this and highlight the practical application of our HI-Mol framework, we further show the model's capability to generate molecules with the desired PLogP. In Figure 3, HI-Mol generates realistic molecules with the target PLogP, even when the desired condition $\gamma$ is unseen in the training molecules. The overall results show that our HI-Mol exhibits a huge potential for real-world scenarios where we aim to generate molecules with a specific target property.

**Low-shot molecular property prediction.** We show that the generated molecules by HI-Mol can be utilized to improve the performance of classifiers for low-shot molecular property prediction. Here, we collect low-shot molecules from the MoleculeNet benchmark (Wu et al., 2018) and generate molecules via molecular generative models for each label. In Table 5, HI-Mol consistently shows the superior $\Delta$ROC-AUC[5] score. This demonstrates the efficacy of HI-Mol to learn the concept, e.g., activeness and in-activeness, of each label information with a limited number of molecules. In practical scenarios, where the label information is hard to achieve, our HI-Mol indeed plays an important role in improving the classifier. We provide experimental details in Appendix L.

---

[5]This score is calculated by the improvement of the ROC-AUC score when the generated molecules are additionally added to the original low-shot training data; higher is better.

Table 4: Results of molecular property maximization task. We report the top-3 property scores denoted by 1st, 2nd, and 3rd. The baseline scores are drawn from Ahn et al. (2022).

| Method | PlogP | | |
| --- | --- | --- | --- |
| | 1st | 2nd | 3rd |
| GVAE (Kusner et al., 2017) | 2.94 | 2.89 | 2.80 |
| SD-VAE (Dai et al., 2018) | 4.04 | 3.50 | 2.96 |
| JT-VAE (Jin et al., 2018) | 5.30 | 4.93 | 4.49 |
| MHG-VAE (Kajino, 2019) | 5.56 | 5.40 | 5.34 |
| GraphAF (Shi et al., 2020) | 12.23 | 11.29 | 11.05 |
| GraphDF (Luo et al., 2021) | 13.70 | 13.18 | 13.17 |
| STGG (Ahn et al., 2022) | 23.32 | 18.75 | 16.50 |
| **HI-Mol (Ours; 1%)** | **24.67** | **21.72** | **20.73** |

Table 5: Average $\Delta$ROC-AUC of the low-shot property prediction tasks with 20 random seeds.

| Dataset | Method | 16-shot | 32-shot |
| --- | --- | --- | --- |
| HIV | DiGress (Vignac et al., 2023) | -2.30 | -2.67 |
| | MiCaM (Geng et al., 2023) | 1.02 | 0.69 |
| | STGG (Ahn et al., 2022) | 0.53 | -0.47 |
| | **HI-Mol (Ours)** | **2.35** | **2.16** |
| BBBP | DiGress (Vignac et al., 2023) | 1.73 | 0.97 |
| | MiCaM (Geng et al., 2023) | 1.91 | 1.78 |
| | STGG (Ahn et al., 2022) | 1.85 | 1.76 |
| | **HI-Mol (Ours)** | **2.73** | **2.64** |
| BACE | DiGress (Vignac et al., 2023) | -0.60 | -0.91 |
| | MiCaM (Geng et al., 2023) | -0.65 | -1.11 |
| | STGG (Ahn et al., 2022) | 2.34 | 2.01 |
| | **HI-Mol (Ours)** | **3.53** | **3.39** |

Table 6: Ablation of the components of hierarchical textual inversion on the QM9 dataset (Ramakrishnan et al., 2014) with 2% subset. We report the results using 10,000 sampled molecules.

| Training prompt | FCD $\downarrow$ | NSPDK $\downarrow$ | Valid. $\uparrow$ | Unique. $\uparrow$ | Novelty $\uparrow$ |
| --- | --- | --- | --- | --- | --- |
| The molecule is a $[S^*]$ | 7.913 | 0.041 | **96.2** | 19.3 | 39.5 |
| The molecule is a $[S^*][D_n^*]$ | 0.486 | 0.002 | 93.8 | 70.8 | 72.3 |
| The molecule is a $[S^*][I_{c_n}^*][D_n^*]$ | **0.434** | **0.001** | 90.7 | **75.8** | **73.5** |

## 4.4 ANALYSIS

**Effect of intermediate tokens.** Recall that we have introduced intermediate text tokens $\{[I_k^*]\}_{k=1}^K$, which are selected in an unsupervised manner during the hierarchical textual inversion to learn some of the cluster-wise features included in given molecules. To validate the effect of our text token design, we visualize the clustering results in Figure 2 by providing groups of the molecules that have the same intermediate token. As shown in this figure, molecules are well grouped according to their common substructures, e.g., a long carbon chain or sulfonyl benzene groups. Such a learning of cluster-wise low-level semantics is indeed beneficial in molecular generation, since molecules often share the concept, e.g., molecular property, even when they have large structural differences.

**Ablation on hierarchical tokens.** To validate the effect of each token in our proposed hierarchical textual inversion, we perform an ablation study by comparing the results with our method where some of the tokens are excluded from the overall framework. Specifically, we compare the generation performance of the following three variants: (1) using the shared token $[S^*]$ only, (2) using $[S^*]$ and the detail tokens $[D_n^*]$, and (3) using all three types of tokens (HI-Mol). Note that for (1), it is impossible to apply our interpolation-based sampling; hence, we use temperature sampling instead based on the categorical distribution from a molecular language model with temperature $\tau = 2.0$. We provide this result in Table 6: as shown in this table, introducing each of the additional tokens successively improves most of the metrics, while maintaining the Validity score as well.

## 5 CONCLUSION

We propose HI-Mol, a data-efficient molecular generation framework that utilizes a molecule-specialized textual inversion scheme. Specifically, we propose to capture the hierarchical information of molecular data in the inversion stage, and use it to sample novel molecules. We hope our method initiates under-explored but crucial research direction in the data-efficient generation of molecules.

**Limitation and future work.** In this work, we apply our novel textual inversion scheme to the molecular language model (Edwards et al., 2022), where developing such a model is a very recently considered research direction. An important future work would be improving the large-scale molecular language models themselves, e.g., the breakthroughs in the image domain (Rombach et al., 2022), which will allow more intriguing applications of HI-Mol, such as composition (see Appendix F).

ETHICS STATEMENT

This work will facilitate research in molecular generation, which can speed up the development of many important generation tasks such as finding drugs for a specific organ and disease when the hit molecules are rarely known. However, malicious use of well-learned molecular generative model poses a potential threat of creating hazardous molecules, such as toxic chemical substances. It is an important research direction to prevent such malicious usage of generative models (OpenAI, 2023). On the other hand, molecular generation is also essential for generating molecules to defend against harmful substances, so the careful use of our work, HI-Mol, can lead to more positive effects.

REPRODUCIBILITY STATEMENT

We provide explicit description of our training objective and the sampling method in Section 3.3. We list the hyper-parameter information and the hardware information in Appendix A. We describe the details of datasets and evaluation metrics in Appendix B and C, respectively. We provide our molecule modification algorithm in Appendix H. We submit the code implementation of our HI-Mol framework as a supplementary material.

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

# Appendix: Data-Efficient Molecular Generation with Hierarchical Textual Inversion

## A  METHOD DETAILS

We utilize a recently introduced text-to-molecule model, MolT5-Large-Caption2Smiles (Edwards et al., 2022) in our HI-Mol framework.[6] This model is constructed upon a text-to-text model, T5 (Raffel et al., 2020), and molecular information is injected by additional training with both unpaired SMILES (Weininger, 1988) string and caption-SMILES paired dataset. Our experiment is conducted for 1,000 epochs using a single NVIDIA GeForce RTX 3090 GPU with a batch size of 4. We use AdamW optimizer with $\epsilon = 1.0 \times 10^{-8}$ and let the learning rate $0.3$ with linear scheduler. We clip gradients with the maximum norm of 1.0. We update the assigned cluster $c_n$ of each molecule for the first 5 epochs following Eq. (1). For interpolation-based sampling, we choose a uniform distribution $p(\lambda)$, (i.e., $p(\lambda) \coloneqq \mathcal{U}(l, 1 - l)$), where $\lambda$ controls relative contributions of interpolated token embeddings. We set $l = 0.0$ on the datasets in MoleculeNet benchmark (Wu et al., 2018), and $l = 0.3$ on the QM9 dataset (Ramakrishnan et al., 2014).

## B  DATASETS

**MoleculeNet dataset.** We perform generation experiments on single-task datasets, HIV, BBBP, and BACE, from MoleculeNet (Wu et al., 2018) benchmark. For each dataset, molecules are labeled with 0 or 1, based on its activeness of the target property:

- *HIV* consists of molecules and its capability to prevent HIV replication.
- *BBBP* consists of molecules and whether each compound is permeable to the blood-brain barrier.
- *BACE* consists of molecules and its binding results for a set of inhibitors of $\beta$-secretase-1.

We collect active (e.g., label-1) molecules to train molecular generative models. We utilize a common splitting scheme for MoleculeNet dataset, *scaffold split* with split ratio of train:valid:test = 80:10:10 (Wu et al., 2018). We emphasize that such *scaffold split* is widely considered in molecular generation domain (Ahn et al., 2022). Additional statistics for datasets on MoleculeNet are provided in Table 7.

Table 7: MoleculeNet downstream classification dataset statistics

| Dataset | HIV | BBBP | BACE |
|---|---|---|---|
| Number of molecules | 41,127 | 2,039 | 1,513 |
| Number of active molecules | 1,443 | 1,567 | 691 |
| Avg. Node | 25.51 | 24.06 | 34.08 |
| Avg. Degree | 54.93 | 51.90 | 73.71 |

**QM9 dataset.** We perform generation experiments on the QM9 dataset (Ramakrishnan et al., 2014), which is a widely adopted to benchmark molecular generation methods. This dataset consists of 133,885 small orginic molecules. We follow the dataset splitting scheme of (Ahn et al., 2022) and randomly subset the training split with 2%, 5%, 10%, and 20% ratio for training our HI-Mol.

---

[6]https://huggingface.co/laituan245/molt5-large-caption2smiles

## C  EVALUATION METRICS

We mainly utilize 6 metrics to incorporate diverse aspects for evaluation of the generated molecules. We adopt 5 metrics (FCD, NSPDK, Validity, Uniqueness, Novelty) used in (Jo et al., 2022):

- **Fréchet ChemNet Distance (FCD)** (Preuer et al., 2018) evaluates the distance between the generated molecules and test molecules using the activations of the penultimate layer of the ChemNet, similar to popular Fréchet inception distance (FI) used in image domain (Heusel et al., 2017):

$$\mathrm{FCD} := \|m - m_g\|_2^2 + \mathrm{Tr}\big(C + C_g - 2(CC_g)^{1/2}\big), \tag{3}$$

  where $m, C$ are the mean and covariance of the activations of the test molecules, and $m_g, C_g$ are the mean and covariance of the activations of the generated molecules.
- **Neighborhood Subgraph Pairwise Distance Kernel MMD (NSPDK)** (Costa & De Grave, 2010) calculates the maximum mean discrepancy between the generated molecules and test molecules. We follow the evaluation protocol in (Jo et al., 2022), to incorporate both atom and bond features.
- **Validity (Valid.)** is the ratio of the generated molecules that does not violate chemical validity, e.g., molecules that obey the valency rule.
- **Uniqueness (Unique.)** is the ratio of different samples over total valid generated molecules.
- **Novelty** is the ratio of valid generated molecules that are not included in the training set.

We introduce an additional metric (Active ratio) to evaluate how the generated molecules are likely to be active, e.g., label-1 on our target property:

- **Active ratio (Active.)** is the ratio of the valid generated molecules that are active.

We utilize pre-trained classifiers to measure the activeness of the generated molecules. To be specific, we train a graph isomorphism network (GIN, Xu et al., 2019a) with the entire training split, e.g., contains both active (label-1) and inactive (label-0) molecules, of each dataset in the MoleculeNet benchmark (Wu et al., 2018). We train 5-layer GIN with a linear projection layer for 100 epochs with Adam optimizer, a batch size of 256, a learning rate of 0.001, and a dropout ratio of 0.5. We select the classifier of the epoch with the best validation accuracy. The accuracies of the pre-trained classifier on the validation split are 98.2%, 86.3%, and 86.1%, respectively. We calculate Active ratio by the ratio of the generated molecules that this classifier classifies as label-1.

# D    BASELINES

In this paper, we compare our method with an extensive list of baseline methods in the literature of molecular generation. We provide detailed descriptions of the baselines we considered:

- **GDSS** (Jo et al., 2022) proposes a diffusion model for graph structure, jointly learning both node and adjacency space by regarding each attributes as continuous values.
- **DiGress** (Vignac et al., 2023) proposes a discrete diffusion process for graph structure to properly consider categorical distributions of node and edge attributes.
- **DEG** (Guo et al., 2022) suggests constructing molecular grammars from automatically learned production rules for data-efficient generation of molecules. Due to the high computational complexity of the grammar construction, this method can only be applied to structurally similar molecules, e.g., monomers or chain-extenders, with an extremely limited number of molecules (∼100 molecules with high structural similarity). Nevertheless, we compare with this method in the extremely limited data regime of Appendix F.
- **JT-VAE** (Jin et al., 2018) proposes a variational auto-encoder that represents molecules as junction trees, regarding motifs of molecules as the nodes of junction trees.
- **PS-VAE** (Kong et al., 2022) utilizes a principal subgraph as a building block of molecules and generates molecules via merge-and-update subgraph extraction.
- **MiCaM** (Geng et al., 2023) introduces a connection-aware motif mining method to model the target distribution with the automatically discovered motifs.
- **CRNN** (Segler et al., 2018) builds generative models of SMILES strings with recurrent decoders.
- **STGG** (Ahn et al., 2022) introduces a spanning tree-based molecule generation which learns the distribution of intermediate molecular graph structure with tree-constructive grammar.
- **GraphAF** (Shi et al., 2020) proposes an auto-regressive flow-based model for graph generation.
- **GraphDF** (Luo et al., 2021) introduces an auto-regressive flow-based model with discrete latent variables.
- **MoFlow** (Zang & Wang, 2020) utilizes a flow-based model for one-shot molecular generation.
- **EDP-GNN** (Niu et al., 2020) proposes a one-shot score-based molecular generative model, utilizing a discrete-step perturbation procedure of node and edge attributes.
- **GraphEBM** (Liu et al., 2021) introduces a one-shot energy-based model to generate molecules by minimizing energies with Langevin dynamics.
- **GSDM** (Luo et al., 2022) is a follow-up work of GDSS (Jo et al., 2022), suggesting to consider the spectral values of adjacency matrix instead of adjacency matrix itself.
- **CG-VAE** (Liu et al., 2018) proposes a recursive molecular generation framework that generates molecules satisfying the valency rules by masking out the action space.

# E ABLATION STUDY

Table 8: Ablation on the text prompts for interpolation-based sampling on the 2% subset of QM9.

| Generation prompt | FCD $\downarrow$ | NSPDK $\downarrow$ | Valid. $\uparrow$ | Unique. $\uparrow$ | Novelty $\uparrow$ |
|---|---|---|---|---|---|
| The molecule is a $[S^*][I_{c_n}^*][D_n^*]$ | **0.210** | **0.001** | **92.2** | 61.4 | 47.5 |
| The molecule is similar to $[S^*][I_{c_n}^*][D_n^*]$ | 0.234 | **0.001** | 91.1 | 63.4 | 50.6 |
| A similar molecule of $[S^*][I_{c_n}^*][D_n^*]$ | 0.271 | **0.001** | 91.5 | 65.0 | 52.6 |
| The chemical is similar to $[S^*][I_{c_n}^*][D_n^*]$ | 0.437 | 0.002 | 90.2 | 75.5 | 72.4 |
| A similar chemical of $[S^*][I_{c_n}^*][D_n^*]$ | 0.434 | **0.001** | 90.7 | **75.8** | **73.5** |

Table 9: Ablation on the hierarchical tokens on the 2% subset of QM9.

| Training prompt | FCD $\downarrow$ | NSPDK $\downarrow$ | Valid. $\uparrow$ | Unique. $\uparrow$ | Novelty $\uparrow$ |
|---|---|---|---|---|---|
| The molecule is a $[S_1^*][S_2^*][S_3^*]$ | 6.529 | 0.032 | **96.6** | 21.4 | 37.2 |
| The molecule is a $[S_1^*][S_2^*][D_n^*]$ | 0.474 | 0.002 | 87.0 | 72.9 | 72.0 |
| The molecule is a $[S_1^*][I_{c_n}^*][D_n^*]$ | **0.434** | **0.001** | 90.7 | **75.8** | **73.5** |

Table 10: Ablation on the number of clusters $K$ in Eq. (1) on the 2% subset of QM9.

| K | FCD $\downarrow$ | NSPDK $\downarrow$ | Valid. $\uparrow$ | Unique. $\uparrow$ | Novelty $\uparrow$ |
|---|---|---|---|---|---|
| 0 | 0.486 | 0.002 | 93.8 | 70.8 | 72.3 |
| 1 | 0.474 | 0.002 | 87.0 | 72.9 | 72.0 |
| 3 | 0.455 | 0.002 | 88.9 | 76.5 | 71.1 |
| 5 | 0.443 | **0.001** | 88.0 | 77.0 | 73.2 |
| 10 | 0.434 | **0.001** | **90.7** | 75.8 | 73.5 |
| 20 | **0.430** | **0.001** | 87.9 | **77.3** | 73.8 |
| 30 | 0.436 | **0.001** | 88.9 | 77.2 | **73.9** |
| 2,113 | 0.443 | **0.001** | 86.2 | 75.4 | 72.6 |

**Effect of prompt.** In Table 8, we show the ablation results on the generation prompt for embedding interpolation-based sampling. We observe that we obtain low FCD and NSPDK scores when we use a prompt similar to the training prompt. However, such choices yield low Novelty scores, generating the many molecules contained in the training samples. The prompt we utilize generates more novel molecules while preserving the state-of-the-art FCD and NSPDK scores.

**Effect of hierarchical tokens.** In Table 9, we additionally conduct an ablation study on the effect of the hierarchical tokens. We compare our design with different choice of hierarchy: (1) utilization of only shared tokens, and (2) utilization of shared and detail tokens (without intermediate tokens). For (1), we use temperature sampling instead based on the categorical distribution from a molecular language model with temperature $\tau = 2.0$ since it is impossible to apply our interpolation-based sampling. The results show that consideration of each shared, intermediate, and detail tokens is indeed important for improving the quality measured with various metrics.

**Effect of $K$.** In Table 10, we report the quantitative results of the following cases. First, we consider our proposed design with varying $K$ from 3 to 30. In addition, we consider three other designs that do not contain intermediate tokens to verify the effect of them: (a) $[S_1^*][D_n^*]$ that the intermediate tokens are removed, i.e., $K$=0, (b) $[S_1^*][S_2^*][D_n^*]$ that the intermediate tokens are replaced with a shared token $[S_2^*]$, i.e., $K$=1, and (c) $[S^*][D_{1,n}^*][D_{2,n}^*]$ that the intermediate tokens are replaced with a detail token $[D_{1,n}^*]$, i.e., $K$=2,113. The results exhibit that the intermediate tokens are indeed crucial for the performance, given that the performance $10 \leq K \leq 30$ is much better than (a), (b) and (c). We find that the overall performance is rather degraded with $K$=2,113 compared to $K$=10, 20, and 30. We hypothesis that this is because the sharing of the coarse-grained common features (i.e., intermediate tokens) serves to regularize the fine-grained features (i.e., detail tokens) which are biased toward a single molecule in the embedding interpolation-based sampling. We also remark that we did not put much effort on tuning $K$, e.g., $K$=20 improves FCD as $0.434 \to 0.430$ from $K$=10.

# F   ADDITIONAL EXPERIMENTS

Table 11: Generated molecules from HI-Mol with compositional prompt. We invert 4 aromatic molecules (top row) with the prompt "The molecule is a $[S^*][D_i^*]$". With learned embeddings of $[S^*]$ and $[D_i^*]$, we generate molecules (bottom row) with "The molecule is a boron compound of $[S^*][\bar{D}^*]$". We circle the substructures which indicate that the generated molecules indeed satisfy the condition of the given language prompt.

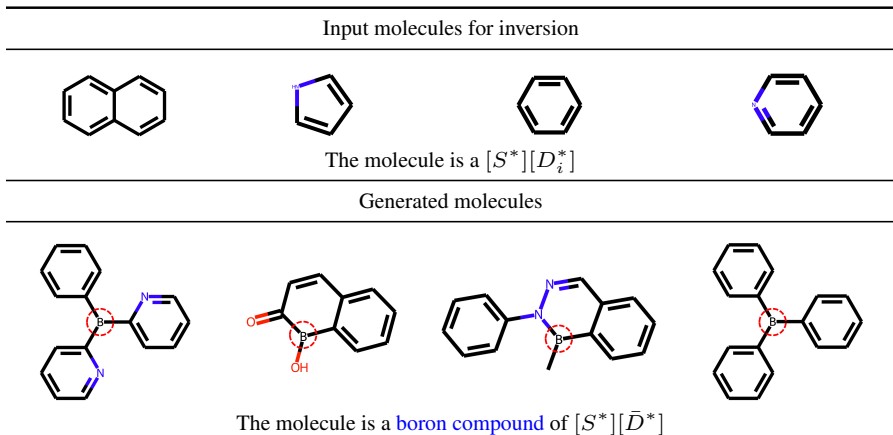

Table 12: Results on (1) learning several concepts (the first row) and (2) learning an underlying concept among diverse molecules (the second row).

|  | MiCaM | STGG | GSDM | HI-Mol (Ours) |
|---|---|---|---|---|
| Success ratio (%) | 18.2 | 33.2 | 0.0 | **52.0** |
| Average QED | 0.555 | 0.558 | 0.090 | **0.581** |

**Compositionality.** In Table 11, we explore the compositionality of the learned token embeddings from HI-Mol. We learn the common features of 4 aromatic molecules[7], e.g., naphthalene, pyrrole, benzene, and pyridine, via textual inversion. Then, we generate molecules with an additional condition via language prompt. We observe that the generated molecules both satisfy (1) the learned common concept of aromatic molecules and (2) the additional conditions from the language prompt. Although our current molecular language model (Edwards et al., 2022) shows some interesting examples of composition between natural language and the learned concept, we strongly believe that future advances in molecular language models will provide more intriguing examples in this application.

**Learning complex molecular concepts.** In this section, we explore the ability of HI-Mol to learn more complex molecular concepts. We conduct two kinds of experiments. Firstly, we impose several target concepts for molecular generation. We collect 300 molecules from GuacaMol (Brown et al., 2019) which satisfy QED>0.5, SA>2.5, and GSK3B>0.3.[8] With these molecules, we check whether the generative models can learn to model several molecular concepts. We report the ratio of the generated molecules that satisfy the aforementioned condition, e.g., QED>0.5, SA>2.5, and GSK3B>0.3, as the Success ratio in Table 12. Our HI-Mol shows superior results on learning several concepts, e.g., $33.2 \rightarrow 52.0$, compared to the most competitive baseline, STGG (Ahn et al., 2022). Secondly, we explore whether HI-Mol can learn the "underlying" molecular property, e.g., QED, among structurally diverse molecules. We curate 329 molecules in the QM9 dataset (Ramakrishnan et al., 2014) where (a) each molecule in this subset has a Tanimoto similarity of no higher than 0.4 with any other molecule in the subset and (b) all the molecules in this subset have a high QED ratio greater than 0.6. The average QED in Table 12 shows that HI-Mol generates molecules with high QED even when the training molecules are structurally largely different, i.e., HI-Mol indeed learns the underlying molecular concept.

---

[7]These molecules share several chemical properties such as resonance and planar structure.
[8]QED, SA, and GSK3B measure the drug-likeness, synthesizability, activity to GSK3B, respectively.

Table 13: Quantitative results of the few-shot generation experiments on subsets of the HIV dataset (Wu et al., 2018). We generate the same number of molecules as the number of the training samples. Due to the large training cost, we report the score of DEG (Guo et al., 2022) only for 30 samples.

| # Samples | Method | Class | Grammar | Active. ↑ | FCD ↓ | NSPDK ↓ | Valid. ↑ | Unique. ↑ | Novelty ↑ |
|---|---|---|---|---|---|---|---|---|---|
| 30 | DEG (Guo et al., 2022) | Graph | ✓ | 3.3 | 39.2 | 0.105 | **100** | **100** | **100** |
| | STGG (Ahn et al., 2022) | SMILES | ✓ | 0.0 | 41.5 | 0.110 | **100** | 67 | **100** |
| | CRNN (Segler et al., 2018) | SMILES | ✗ | 0.0 | 40.0 | 0.121 | 80 | 71 | **100** |
| | **HI-Mol (Ours)** | SMILES | ✗ | **8.3** | **34.8** | **0.103** | 80 | 75 | **100** |
| 150 | STGG (Ahn et al., 2022) | SMILES | ✓ | 1.3 | 28.2 | 0.054 | **100** | 90 | **100** |
| | CRNN (Segler et al., 2018) | SMILES | ✗ | 1.3 | 30.1 | 0.063 | 50 | 84 | **100** |
| | **HI-Mol (Ours)** | SMILES | ✗ | **8.3** | **22.1** | **0.038** | 64 | **91** | **100** |
| 500 | STGG (Ahn et al., 2022) | SMILES | ✓ | 1.3 | 22.8 | 0.041 | **100** | 74 | **100** |
| | CRNN (Segler et al., 2018) | SMILES | ✗ | 2.7 | 30.0 | 0.064 | 51 | **100** | **100** |
| | **HI-Mol (Ours)** | SMILES | ✗ | **10.3** | **20.8** | **0.020** | 63 | 91 | **100** |

Table 14: Comparison with pre-trained model of STGG (Ahn et al., 2022) on the HIV dataset.

| Method | Active. ↑ | FCD ↓ | NSPDK ↓ | Valid. ↑ | Unique. ↑ | Novelty ↑ |
|---|---|---|---|---|---|---|
| STGG (from scratch) | 1.6 | 20.2 | 0.033 | **100** | **95.8** | **100** |
| STGG (fine-tuned) | 3.6 | 20.0 | 0.030 | **100** | 87.1 | **100** |
| **HI-Mol (Ours)** | **11.4** | **16.6** | **0.019** | **100** | 95.6 | **100** |

Table 15: Comparison with the large-scale text-to-molecule models on the HIV dataset.

| Method | Active. ↑ | FCD ↓ | NSPDK ↓ | Valid. ↑ | Unique. ↑ | Novelty ↑ |
|---|---|---|---|---|---|---|
| MolT5-Large-Caption2Smiles (prompting) | 0.0 | 60.6 | 0.196 | **100** | 14.6 | **100** |
| Text+Chem T5-augm-base (prompting) | 0.0 | 62.2 | 0.188 | **100** | 90.2 | **100** |
| MolT5-Large-Caption2Smiles (fine-tuned) | 6.0 | 23.3 | 0.024 | **100** | **96.0** | **100** |
| Text+Chem T5-augm-base (fine-tuned) | 5.8 | 22.4 | 0.023 | **100** | 95.4 | **100** |
| **HI-Mol (Ours)** | **11.4** | **16.6** | **0.019** | **100** | 95.6 | **100** |

**Extremely limited data regime.** Since our model exploits the power of large molecular language models by designing a molecule-specialized textual inversion scheme, one can expect our model to be beneficial in extremely limited data regimes compared with prior methods. To verify this, we conduct an experiment using only a subset of the HIV dataset and report its quantitative result in Table 13. Even with this situation, HI-Mol still outperforms prior state-of-the-art molecular generation methods, e.g., our method improves FCD as 39.2 → 34.8 when trained with 30 samples.

**Comparison with pre-trained model.** In Table 14, we report the performance of the baseline method by fine-tuning the pre-trained baseline model. Specifically, we fine-tune the model of STGG (Ahn et al., 2022) pre-trained with the ZINC250k dataset (Irwin et al., 2012) on the HIV dataset (Wu et al., 2018). We observe that HI-Mol still achieves significantly better performance in overall metrics, e.g., 20.0 → 16.6 and 0.030 → 0.019 in FCD and NSPDK, respectively.

**Comparison with frozen large-scale text-to-molecule models.** In Table 15, we report the performance of the large-scale text-to-molecule models with (1) prompting and (2) fine-tuning. We first remark that it is not feasible to prompt all the training molecules to the large-scale text-to-molecule models due to the maximum token length of the input prompt. Specifically, the maximum token length of the text-to-molecule model is only 1024 and 512 for Edwards et al. (2022) and Christofidellis et al. (2023), respectively, while we have thousands of molecules and each SMILES representation of a molecule sometimes consists of more than 100 tokens. Nevertheless, we provide the molecular generation results of the text-to-molecule models with the text prompt describing the task of the HIV dataset.[9] For fine-tuning experiments, we fine-tune the text-to-molecule models with the entire dataset. We remark that fine-tuning with low-shot examples is known to lead suboptimal performance (Mo et al., 2020; Zhao et al., 2020; Moon et al., 2022) in various domains. For text-to-molecule models, we apply temperature sampling with $\tau$=2.0 and the modification algorithm in Appendix H. The table below shows that our method outperforms the naïve prompting or fine-tuning of the text-to-molecule models. This confirms that our improved performance is not just due to the large-scale text-to-molecule model, but rather to our carefully designed textual inversion framework.

---

[9]We utilize the official description from the MoleculeNet benchmark. http://moleculenet.org/

# G   DETAILS ON QM9 EXPERIMENTS

Table 16: Qualitative results for molecular generation varying the data ratio on QM9.

| Ratio (%) | Grammar | FCD ↓ | NSPDK ↓ | Valid. ↑ | Unique. ↑ | Novelty ↑ |
|---|---|---|---|---|---|---|
| 2 | ✗ | 0.434 | **0.001** | 90.7 | 75.8 | **73.5** |
|   | ✓ | 0.430 | **0.001** | **100** | 76.1 | 75.6 |
| 5 | ✗ | 0.412 | **0.001** | 89.4 | 85.8 | 70.4 |
|   | ✓ | 0.410 | **0.001** | **100** | 86.4 | 72.4 |
| 10 | ✗ | 0.400 | 0.002 | 87.6 | 87.6 | 71.2 |
|    | ✓ | 0.398 | **0.001** | **100** | 88.3 | 73.2 |
| 20 | ✗ | 0.384 | **0.001** | 86.7 | 87.8 | 70.0 |
|    | ✓ | **0.383** | **0.001** | **100** | **88.7** | 71.8 |

Table 17: Comparison with the baseline with high Novelty via resampling strategy on QM9.

| Method | Resampling ratio | FCD ↓ | NSPDK ↓ | Valid. ↑ | Unique. ↑ | Novelty ↑ |
|---|---|---|---|---|---|---|
| GDSS (Jo et al., 2022) | 1.0 | 2.900 | 0.003 | 95.7 | 98.5 | 86.3 |
| **HI-Mol (Ours; 2%)** | 1.9 | **0.601** | **0.002** | **100** | **100** | **100** |

In Table 16, we report additional experimental results varying the data ratio from 2% (2,113 molecules) to 20% (21,126 molecules). In particular, when we use 20% of the training data the performance improves further by $0.430 \rightarrow 0.383$ (compared to using 2% of training data), i.e., our HI-Mol better learns molecule distribution when more molecules are available for training.

We note that there is a fundamental trade-off between FCD and Novelty. If the generated molecules have many overlaps with training molecules, i.e., low Novelty, the FCD score improves, i.e., decreases, since the generated molecules are more likely to follow the target distribution. Therefore, it is crucial to compare FCD under a similar Novelty score. Therefore, in Table 17, we report the generation results with the resampling strategy, i.e., we sample molecules until we have 10,000 molecules with Validity, Uniqueness, and Novelty scores as 100 and we reject samples that violate these scores. We denote the relative ratio of the total sampling trial (including the rejected ones) as Resampling ratio. Here, we remark that such resampling process does not incur much computational cost, e.g., only 1.8 sec for a sample (see Appendix J for analysis on time complexity). The result shows that HI-Mol generates high-quality novel molecules from our desired target distribution.

# H   MODIFICATION ALGORITHM

---

**Algorithm 1:** Modification algorithm for an invalid SMILES string

---

**Input:** An invalid SMILES string
**Output:** A modified SMILES string

1 **while** *exist a branch closing token token prior to a branch opening token* **do**
2   │ Remove the corresponding branch closing token.            // "CC)CCC" to "CCCCC"
3 **while** *exist an unclosed branch opening token* **do**
4   │ Add the the branch closing token at the end of the string.   // "CC(CCC" to "CC(CCC)"
5 **while** *exist an unclosed ring opening token* **do**
6   │ Remove the ring opening token.                          // "CC1CCC" to "CCCCC"
7 **while** *exist an atom that exceeds the valency* **do**
8   │ Randomly drop a branch to satisfy the valency.          // "C#C(=CC)C to "C#CC"
9 **while** *exist a ring with less than 3 atoms* **do**
10  │ Remove the ring opening/closing token.                  // "CC1C1 to "CCC"

---

# I ANALYSIS ON INTERPOLATION-BASED SAMPLING

Table 18: Generated molecules from HI-Mol with varying $\lambda$ in Eq. (2). Samples are generated with the prompt "A similar chemical of $[S^*][\bar{I}^*][\bar{D}^*]$". The columns $[D_i^*]$ and $[D_j^*]$ denote molecules in the HIV dataset (Wu et al., 2018) whose token embeddings are interpolated for each row.

| $[D_i^*]$ | A similar chemical of $[S^*][I^*][\bar{D}^*]$ | | | | | $[D_j^*]$ |
|---|---|---|---|---|---|---|
| | $\lambda = 0.0$ | $\lambda = 0.3$ | $\lambda = 0.5$ | $\lambda = 0.7$ | $\lambda = 1.0$ | |
| | $\lambda = 0.0$ | $\lambda = 0.3$ | $\lambda = 0.5$ | $\lambda = 0.7$ | $\lambda = 1.0$ | |

Table 19: Generated molecules from HI-Mol with varying $\lambda$ in Eq. (2). We interpolate a single-level token, e.g., "A similar chemical of $[S^*][\bar{I}^*][D^*]$" and "A similar chemical of $[S^*][I^*][\bar{D}^*]$".

A similar chemical of $[S^*][\bar{I}^*][D^*]$

| $\lambda = 0.0$ | $\lambda = 0.3$ | $\lambda = 0.5$ | $\lambda = 0.7$ | $\lambda = 1.0$ |
|---|---|---|---|---|

A similar chemical of $[S^*][I^*][\bar{D}^*]$

| $\lambda = 0.0$ | $\lambda = 0.3$ | $\lambda = 0.5$ | $\lambda = 0.7$ | $\lambda = 1.0$ |
|---|---|---|---|---|

Note that our sampling is based on the interpolation of two different token embeddings with different values of $\lambda \sim p(\lambda)$. In Table 18, we provide how the generated molecules are changed with different values of $\lambda$. With varying $\lambda$, one can observe that the generated molecules (1) maintain some original important low-level semantics and (2) introduce some novel aspects distinct from both original semantics. For example, $\lambda = 0.7$ in the first row of Table 18 introduces a new 4-membered ring system while preserving the phosphorous-sulfur double bond structure of the original features in $[D_j^*]$. This observation exhibits that our embedding space models the manifold of underlying target distribution effectively, enabling data-efficient sampling from the target distribution. We also provide the generated samples from different hierarchies. Interpolating intermediate tokens (see the first row of Table 19) change the low-level semantics, i.e., size of molecules, of the generated molecules and interpolating detail (see the second row of Table 19) tokens change the high-level features, i.e., insertion of a single atom, of the generated molecules.

# J COMPLEXITY

Table 20: Time and space complexity of each molecular generative method.

| | JT-VAE | PS-VAE | MiCaM | STGG | CRNN | GDSS | GSDM | DiGress | HI-Mol (Ours) |
|---|---|---|---|---|---|---|---|---|---|
| Time complexity (s) | 4.8 | 0.1 | 0.9 | 0.7 | 0.5 | 71.2 | 2.0 | 9.1 | 1.8 |
| Space complexity (GB) | 0.4 | 1.2 | 1.6 | 2.1 | 0.4 | 1.2 | 1.1 | 1.5 | 4.8 |

In Table 20, we provide the time and space complexity to generate a molecule via various molecular generative models. For time complexity, measured with a single RTX 3090 GPU, HI-Mol takes about 1.8 seconds to sample a single molecule, while other methods, e.g., GDSS and DiGress, require more time due to denoising diffusion steps. For memory complexity, HI-Mol requires 4.8GB of GPU VRAM space due to the usage of the large model. We believe that reducing this space for large language models, e.g., through Dao et al. (2022) will be an interesting future direction.

## K    DISCUSSION ON MOLECULAR OPTIMIZATION

In Table 4, we have shown the usefulness of our HI-Mol to maximize the PLogP value of the generated molecules. While this evaluation setup for molecular optimization is a common and popular choice in molecular domain (Jin et al., 2018; Shi et al., 2020; Luo et al., 2021; Ahn et al., 2022), some prior works have noted that solely maximizing the PLogP value may yield unstable or hard-to-synthesize molecules (Gao & Coley, 2020; Coley, 2021; Ahn et al., 2022). In Figure 4, we show the visualizations of the optimized molecules with the highest PLogP values. Similar to the most competitive baseline, STGG (Ahn et al., 2022), our optimized molecules contain a large number of atoms, and thus relatively hard to synthesize. Although these results show that our HI-Mol effectively learns

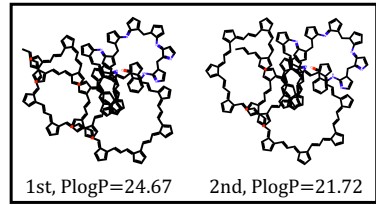

1st, PlogP=24.67        2nd, PlogP=21.72

Figure 4: Visualizations of the generated molecules with $\gamma = 50$. The maximum PLogP among the training molecules is 4.52.

to incorporate the condition PLogP in a data-efficient manner, it would be an important research direction to develop an evaluation framework for molecular optimization that takes into account the "realistic-ness", e.g., stability and synthesizability, of the molecules.

## L    DETAILS ON LOW-SHOT MOLECULAR PROPERTY PREDICTION

Table 21: Results on low-shot classification on the MoleculeNet benchmark. We report the average and 95% confidence interval of the test ROC-AUC scores within 20 random seeds.

| Dataset | Method | 16-shot | 32-shot |
|---------|--------|---------|---------|
| HIV | DiGress (Vignac et al., 2023) | -2.30±3.50 | -2.67±3.15 |
| | MiCaM (Geng et al., 2023) | 1.02±3.29 | 0.69±2.09 |
| | STGG (Ahn et al., 2022) | 0.53±2.79 | -0.47±2.36 |
| | **HI-Mol (Ours)** | **2.35**±2.71 | **2.16**±1.64 |
| BBBP | DiGress (Vignac et al., 2023) | 1.73±1.53 | 0.97±1.99 |
| | MiCaM (Geng et al., 2023) | 1.91±2.13 | 1.78±1.98 |
| | STGG (Ahn et al., 2022) | 1.85±1.83 | 1.76±1.72 |
| | **HI-Mol (Ours)** | **2.73**±2.01 | **2.64**±1.75 |
| BACE | DiGress (Vignac et al., 2023) | -0.60±2.88 | -0.91±1.82 |
| | MiCaM (Geng et al., 2023) | -0.65±3.17 | -1.11±2.95 |
| | STGG (Ahn et al., 2022) | 2.34±2.15 | 2.01±1.45 |
| | **HI-Mol (Ours)** | **3.53**±1.57 | **3.39**±1.80 |

In Table 21, we report the full results of low-shot molecular property prediction experiments with averages and 95% confidence intervals. With randomly sampled low-shot molecules from the train split (used in our main experiments of Table 1), we generate ×3 number of valid molecules via generative models, e.g., we generate 96 molecules for 32-shot experiments. For the classifier, we utilize the 5-layer GIN (Xu et al., 2019a) from You et al. (2020), which is pre-trained with unlabeled molecules via self-supervised contrastive learning. We fine-tune this model for 100 epochs by introducing a linear projection head for each dataset. We use Adam optimizer with a learning rate of 0.0001 and no weight decay. The results are calculated based on the test ROC-AUC score of the epoch with the best validation ROC-AUC score. Specifically, we consider two scenarios: (1) training the classifier with only the low-shot molecules and (2) training the classifier with both the original low-shot molecules and the generated molecules via the molecular generative model. We report ΔROC-AUC score, calculated by the subtraction of the ROC-AUC score of (1) from (2).

