# OpenReview forum: "Data-Efficient Molecular Generation with Hierarchical Textual Inversion"
_ICLR.cc/2024/Conference — Submitted to ICLR 2024_

### Official Review · Reviewer_9Hsy · 2023-10-28

**Soundness:** 3 good
**Presentation:** 3 good
**Contribution:** 3 good
**Rating:** 6
**Confidence:** 4

**Summary:**

Inspired by recent textual inversion technique in the visual domain, the authors proposed Hierarchical textual Inversion for Molecular generation (HI-Mol), a novel data-efficient molecular generation method. Extensive experiments demonstrate the superiority of HI-Mol with notable data-efficiency.

**Strengths:**

1. Adapting textual inversion to the molecule domain is novel.
2. The method is well-introduced and convincing.
3. The authors validated the effectiveness of HI-Mol on several downstream tasks including the molecular optimization for PLogP and the low-shot molecular property prediction on MoleculeNet.

**Weaknesses:**

1. It is worth mentioning some recent works on molecule generation in related works such as:

[1] Hoogeboom E, Satorras V G, Vignac C, et al. Equivariant diffusion for molecule generation in 3d, ICML 2022
[2] Zhang Z, Liu Q, Lee C K, et al. An equivariant generative framework for molecular graph-structure Co-design. Chemical Science 2023
[3] Flam-Shepherd D, Zhu K, Aspuru-Guzik A. Language models can learn complex molecular distributions. Nature Communications, 2022

2. Could HI-Mol leverage the structural information of molecules? Could it be adapted for 3D molecule generation?
3. How to effectively interpret the learned tokens? Do they have chemical meanings?

**Questions:**

N.A.

---

> ### Author Response · Authors · 2023-11-16
> **Response to Reviewer 9Hsy**
>
> Dear Reviewer 9Hsy,
>
> We sincerely appreciate your efforts in reviewing our manuscript. We respond to each comment in the following content. We carefully incorporated the discussions into the revised manuscript. We highlighted the revised contents in "$\text{\color{blue}blue}$" for your convenience to check.
>
> ---
>
> **[W1] It is worth mentioning some recent works on molecule generation in related works [1, 2, 3].**
>
> We appreciate your comment to improve our manuscript. Following your suggestion, we mentioned the related works [1, 2, 3] in Section 2 of our revised manuscript.
>
> ---
> **[W2] Could HI-Mol be adapted for 3D molecule generation by leveraging the structural information?**
>
> We strongly believe that our framework can be adapted for 3D molecular generation by leveraging the 3D structural information of molecules. Although we use the state-of-the-art text-to-molecule model (MolT5), which utilizes the SMILES string as the molecule representation, it is certainly possible to model the 'molecule' part of a text-to-molecule model as 3D molecule (with a recently proposed 3D molecular generative model [1] as a decoder architecture) without any methodological modifications of our method.
>
> ---
> **[W3] How to effectively interpret the learned tokens? Do they have chemical meanings?**
>
> Learned tokens have a chemical meaning (i.e., hierarchical features of molecules). For example, detail tokens represent features of a single molecule in the dataset. These learned tokens are effectively interpreted through visualization. For example, in Figure 2 of the manuscript, one can see that certain intermediate tokens represent molecules with chemical substructures such as long carbon chains or sulfonyl groups.
>
> ---
> [1] Hoogeboom et al., Equivariant Diffusion for Molecule Generation in 3D, ICML 2022.\
> [2] Zhang et al., An Equivariant Generative Framework for Molecular Graph-structure Co-design, Chemical Science 2023.\
> [3] Flam-Shephered et al., Language Models Can Learn Complex Molecular Distribution, Nature Communications 2022.

---

> ### Author Response · Authors · 2023-11-20
> **A Gentle Reminder**
>
> Dear Reviewer 9Hsy,
>
> Thank you very much again for your time and efforts in reviewing our paper.
>
> We kindly remind that we have only three days or so in the discussion period.
>
> We just wonder whether there is any further concern and hope to have a chance to respond before the discussion phase ends.
>
> Many thanks, \
> Authors

---

> > ### Comment · Reviewer_9Hsy · 2023-11-23
> > **Response to rebuttal**
> >
> > Thanks to the authors for the detailed response! Most of my concerns are addressed.
> >
> > Bests,

---

> > > ### Author Response · Authors · 2023-11-23
> > > **Thank you for your response.**
> > >
> > > Dear Reviewer 9Hsy,
> > >
> > > Thank you for your comments and for taking the time to review our manuscript. We are pleased to hear that our replies have resolved your concerns.
> > >
> > > If you have any further comments or suggestions, please let us know. We are committed to improving the quality of our work, and we value your feedback.
> > >
> > > Thank you very much,\
> > > Authors

---

### Official Review · Reviewer_Mfay · 2023-10-31

**Soundness:** 3 good
**Presentation:** 3 good
**Contribution:** 3 good
**Rating:** 6
**Confidence:** 3

**Summary:**

Recently proposed molecular generation methods are mainly trained on task-related data, which are computationally expensive. The authors propose a hierarchical textual inversion method for molecular generation to overcome this issue.

**Strengths:**

1.	Introducing the successful textural inversion methods from the computer vision area into the molecular generation area is a good idea.
2.	The experimental results presented in the paper demonstrate the effectiveness of the proposed method.

**Weaknesses:**

1.	The authors should have a clearer motivation figure in the introduction, which could be specific examples of molecules, to demonstrate that the highly complicated and structured nature of molecules makes it difficult to apply textual inversion directly.
2.	The Molecular language model part in the Section 3.2 Preliminaries should be moved to the Related Work section.
3.	Table 2 should also show the results of HI-Mol without grammar.
4.	In Table 6, Valid decreases as the token hierarchical levels increases. The authors should provide some explanations and solutions for this issue.
5.	I suggest that the authors consider rearranging the positions of the figures and tables as some of them are too far from the references in the paper.

**Questions:**

1.	What is the ‘simple resampling strategy’ mentioned in Section 4.2?

---

> ### Author Response · Authors · 2023-11-16
> **Response to Reviewer Mfay**
>
> Dear Reviewer Mfay,
>
> We sincerely appreciate your efforts in reviewing our manuscript. We respond to each comment in the following content. We carefully incorporated the discussions into the revised manuscript. We highlighted the revised contents in "$\text{\color{blue}blue}$" for your convenience to check.
>
>
> ---
> **[W1] Clear motivation figure to demonstrate highly complicated and structured nature of molecules.**
>
> Thank you for the comment to improve the clarity of our manuscript. Following your suggestion, we added Figure 2 as a motivation figure in the introduction section of our revised manuscript.
>
> ---
> **[W2] The molecular language model part in the Section 3.2 should be moved to the Related Work section.**
>
> We appreciate your suggestion. Following your suggestion, we moved the Molecular language model part to the Related Work section in our revised manuscript.
>
> ---
> **[W3] Results of HI-Mol without grammar.**
>
> Thank you for the suggestion. We moved the QM9 results of HI-Mol without grammar to Table 3 (which is originally placed in Table 15 of Appendix G), in our revised manuscript.
>
> ---
> **[W4] In Table 6, why Valid decreases as the token hierarchical level increases? How to resolve it?**
>
> As the token hierarchy increases, Validity decreases because the model interpolates more levels of embeddings to generate more diverse molecules (measured by Unique. and Novelty), i.e., there is a trade-off between the diversity of generated molecules and their validity. However, we emphasize that an invalid molecule can be easily converted to a valid molecule via a simple modification algorithm (in Appendix H), which actually follows our desired distribution (see improvement in FCD score in the table below).
>
> \begin{array}{l|c|ccccc}
> \hline
> \text{QM9} &\text{Grammar} & \text{FCD $\downarrow$} & \text{NSPDK $\downarrow$} & \text{Valid. $\uparrow$} & \text{Unique. $\uparrow$} & \text{Novelty $\uparrow$} \newline
> \hline
> \text{HI-Mol (Ours)} & \text{X} &  0.434 & 0.001 & 90.7 & 75.8 & 73.5 \newline
> \text{HI-Mol (Ours)} & \text{O} &  \textbf{0.430} & \textbf{0.001} & \textbf{100} & \textbf{76.1} & \textbf{75.6} \newline
> \hline
> \end{array}
>
> ---
> **[W5] Rearrangement of the positions of the figures and tables.**
>
> Thank you for the comment to improve our manuscript. Following your suggestion, we carefully rearranged the positions of the figures and tables in our revised manuscript.
>
> ---
> **[Q1] What is ‘simple resampling strategy’ in Section 4.2?**
>
> Thank you for the opportunity to clarify this point. The ‘simple resampling strategy’ means that we ignore the generated sample when it is (1) invalid, (2) identical to an already generated sample, or (3) identical to a sample in the training dataset. By applying this strategy, we obtain a set of generated samples whose Valid., Unique., and Novelty scores are all 100 (note that sampling costs only 1.8 sec per molecule). We consider this scenario to align the Valid., Unique., and Novelty scores with some methods (e.g., GDSS) for a fair comparison in the FCD score. Even in this scenario, HI-Mol significantly outperforms those methods.

---

> ### Author Response · Authors · 2023-11-20
> **A Gentle Reminder**
>
> Dear Reviewer Mfay,
>
> Thank you very much again for your time and efforts in reviewing our paper.
>
> We kindly remind that we have only three days or so in the discussion period.
>
> We just wonder whether there is any further concern and hope to have a chance to respond before the discussion phase ends.
>
> Many thanks, \
> Authors

---

> > ### Comment · Reviewer_Mfay · 2023-11-23
> >
> > Thank you for your response. I have no further questions, and I will keep my score.

---

> > > ### Author Response · Authors · 2023-11-23
> > > **Thank you for your response.**
> > >
> > > Dear Reviewer Mfay,
> > >
> > > Thank you for your comments and for taking the time to review our manuscript. We are pleased to hear that our replies have resolved your concerns.
> > >
> > > If you have any further comments or suggestions, please let us know. We are committed to improving the quality of our work, and we value your feedback.
> > >
> > > Thank you very much,\
> > > Authors

---

### Official Review · Reviewer_5Up6 · 2023-10-31

**Soundness:** 2 fair
**Presentation:** 2 fair
**Contribution:** 3 good
**Rating:** 6
**Confidence:** 4

**Summary:**

The paper introduces Hierarchical Textual Inversion for Molecular generation (HI-Mol), a novel approach to generate molecular structures efficiently with limited datasets. It leverages a new scheme of textual inversion tailored for molecules, using multi-level tokens to capture hierarchical information. This method enables the generation of molecules with significantly less data, showing superiority in various benchmarks such as the MoleculeNet and QM9 datasets, particularly improving data efficiency and performance in molecular property prediction and optimization tasks

**Strengths:**

- The problem they are trying to tackle is important and interesting.
- The idea looks relatively novel and justified since molecules are constructed of similar smaller components.
- The empirical results are promising.

**Weaknesses:**

- The method is not described clearly and in detail. For instance, in the following paragraph of Eq. 1, it is mentioned that the intermediate tokens are "selected" during training. This is unclear and should be discussed in more detail.

-  Figure 1 is not expressive enough to outline the method.

**Questions:**

- The authors have mentioned that they are using the Caption2Smiles frozen model as the backbone. Can they please share just the frozen model performance on the tasks?

- Sensitivity of the model's performance to the number of k sounds very important. According to Appendix E, table 10, there is no benefit in increasing k to more than 10. Do the authors have a hypothesis for that? Aren't the molecules supposed to have a lot more "clusters", sub-components?

- I'm curious how making this approach multi-modal can be helpful. Could graph embeddings or vision embeddings of the molecules provide any benefit? I'm not a molecular properties expert, but I tried a couple of the figures (table 2 and figure 2) with GPT-4 vision, and it gave meaningful explanations. Have the authors investigated this?

- Can the authors please provide the complete Qm9 results in table 15 of Appendix G? Specifically, what are the results at the 50% and 100% ratios?

---

> ### Author Response · Authors · 2023-11-16
> **Response to Reviewer 5Up6 (1/2)**
>
> Dear Reviewer 5Up6,
>
> We sincerely appreciate your efforts in reviewing our manuscript. We respond to each comment in the following content. We carefully incorporated the discussions into the revised manuscript. We highlighted the revised contents in "$\text{\color{blue}blue}$" for your convenience to check.
>
> ---
> **[W1] Clear description of method, e.g., “selected” in Eq. (1).**
>
> Thank you for the comment to improve the quality of our manuscript. For clarification, to learn the cluster-wise features of a molecular dataset, we propose a strategy to assign an appropriate cluster index to each molecule. Specifically, given a molecule, we assign (i.e., “select’’) the intermediate token $[I_k]$ where $k$ minimizes the reconstruction loss among the cluster indices $[1, K]$. This strategy allows us to cluster similar molecules (see Figure 2) in an unsupervised manner. We carefully revised the manuscript to provide a clear description of our method.
>
> ---
> **[W2] Figure 1 is not expressive enough to outline the method.**
>
> Thank you for the suggestion. Following your comment, we revised Figure 1 to better outline our method.
>
> ---
> **[Q1] Performance of the frozen Caption2Smiles model?**
>
> Following your suggestion, we provide the generation results of the frozen Caption2Smiles model [1] on the HIV dataset, where we utilize the task description of the HIV dataset as the text prompt. The table below shows that our method indeed significantly outperforms naïvely prompting frozen Caption2Smiles model, which implies that our proposed framework is essential in achieving the superior performance. We added these respects of discussion and the experimental details in Appendix F of our revised manuscript.
>
> \begin{array}{l|cccccc}
> \hline
> \text{HIV} & \text{Active. $\uparrow$} & \text{FCD $\downarrow$} & \text{NSPDK $\downarrow$} & \text{Valid. $\uparrow$} & \text{Unique. $\uparrow$} & \text{Novelty $\uparrow$} \newline
> \hline
> \text{MolT5-Large-Caption2Smiles [1]} & 0.0 & 60.6 & 0.196 & \textbf{100} & 14.6 & \textbf{100} \newline
> \hline
> \textbf{HI-Mol (Ours)} & \textbf{11.4} & \textbf{16.6} & \textbf{0.019} & \textbf{100} & \textbf{95.6} & \textbf{100} \newline
> \hline
> \end{array}
>
> ---
> **[Q2] In Table 10, there is no benefit in increasing $K$ more than 10. Do the authors have a hypothesis?**
>
> Thank you for the insightful comment. We kindly remark that some previous works on molecular clustering have shown that a thousand of molecules can fall into $\sim$10 clusters [2], which could be a supporting hypothesis on our finding $K=$ 10. While the number of “clusters’’ of molecules can be defined in various ways, we also hypothesize that choosing a relatively small number of clusters (e.g., $K=$ 10) is sufficient in our framework. This is because the sharing of the coarse-grained common features (i.e., intermediate tokens) serves to regularize the fine-grained features (i.e., detail tokens) which are biased toward a single molecule in the embedding interpolation-based sampling. To verify our hypothesis, we conduct an additional experiment, where we replace the intermediate tokens with additional detail tokens for each molecule (i.e., $K=$ 2,113). The reported result below show that the overall performance is rather degraded with $K=$ 2,113 compared to $K=$ 10, 20, and 30. This verifies our hypothesis and justifies our choice of the number of clusters $K$ in our experiments. We added these respects of discussion in Appendix E of our revised manuscript.
>
> \begin{array}{l|ccccc}
> \hline
> \text{$K$} & \text{FCD $\downarrow$} & \text{NSPDK $\downarrow$} & \text{Valid. $\uparrow$} & \text{Unique. $\uparrow$} & \text{Novelty $\uparrow$} \newline
> \hline
> 10 & 0.434 & \textbf{0.001} & \textbf{90.7} & 75.8 & 73.5 \newline
> 20 & \textbf{0.430} & \textbf{0.001} & 87.9 &\textbf{77.3} & 73.8 \newline
> 30 & 0.436 & \textbf{0.001} & 88.9 & 77.2 & \textbf{73.9} \newline
> \text{2,113} & 0.443 & \textbf{0.001} & 86.2 & 75.4 & 72.6 \newline
> \hline
> \end{array}
>
> ---
> **[Q3] Could graph embeddings or vision embeddings of the molecules provide any benefit?**
>
> Thank you for your insightful comment for our future research direction. We strongly believe that graph embeddings or vision embeddings (such as the suggested GPT-4 vision examples) can be incorporated to improve the controllability of molecular generation of our framework, given some observations in the text-to-image domain that explore such a multi-modal approach to improve the controllability of image generation [3].

---

> ### Author Response · Authors · 2023-11-16
> **Response to Reviewer 5Up6 (2/2)**
>
> ---
> **[Q4] In Table 15, what are the results at the 50% and 100% ratio?**
>
> Thank you for the suggestion to improve our manuscript. First, we would like to emphasize that our target problem is molecular generation in the low-data regime, and our method already outperforms state-of-the-art methods with only 2% of training data of the QM9 dataset. For this reason, we thought that the results using 50% and 100% training data are not that meaningful to show the superiority of our method. Nevertheless, for your information, we are currently running the experiments with 50% and 100% QM9 dataset. We will definitely include these results in our final draft (we hope your understanding that it is costly to train with such a large number of molecules in our local machine).
>
> ---
> [1] Edwards et al., Translation between Molecules and Natural Language, EMNLP 2022\
> [2] Hariharan et al., MultiMCS: A Fast Algorithm for the Maximum Common Substructure Problem on Multiple Molecules, JCIM 2011\
> [3] Rombach et al., High-Resolution Image Synthesis with Latent Diffusion Models, CVPR 2022

---

> ### Author Response · Authors · 2023-11-20
> **A Gentle Reminder**
>
> Dear Reviewer 5Up6,
>
> Thank you very much again for your time and efforts in reviewing our paper.
>
> We kindly remind that we have only three days or so in the discussion period.
>
> We just wonder whether there is any further concern and hope to have a chance to respond before the discussion phase ends.
>
> Many thanks, \
> Authors

---

### Official Review · Reviewer_89Qo · 2023-11-01

**Soundness:** 2 fair
**Presentation:** 3 good
**Contribution:** 2 fair
**Rating:** 5
**Confidence:** 4

**Summary:**

The authors present a hierarchical textual inversion strategy, which uniquely selects low-level tokens for each molecule. Subsequently, molecules are generated through a pre-trained text-to-molecule model by interpolating these tokens. Comprehensive testing showcases the marked data-efficiency and superiority of HI-Mol.

**Strengths:**

1. The tackled problem is both intriguing and holds practical significance.

2. The paper is articulate and systematically presented.

3. The introduction of multi-level token embeddings enhances the textual inversion model.

4. Very strong experiments, which clearly show the superiority of the proposed method.

**Weaknesses:**

1. The main concern I have with this paper is its novelty. While the ideas of multi-level molecule representation and embedding interpolation are well-established in the field, the authors merely integrate them into the newly introduced textual inversion framework. This casts doubts over the paper's genuine novelty and the depth of its technical contribution.

2. The rationale for adopting the textual inversion model appears somewhat nebulous. In my understanding, compared to SMILES, graph representations are generally more adept at modeling molecular structures. Notably, many graph-centric molecular generation methods have already incorporated hierarchical concepts. In their experiments, while the authors argue that HI-Mol surpasses several graph-based models, the underlying reasons remain unelucidated. An elucidation on how the textual inversion model specifically augments the molecular generation task, relative to its graph-based counterparts, would be greatly beneficial.

3. There's an absence of crucial baselines. The authors have chosen to compare their work with two SMILES-based baselines, one based on RNN and the other on spanning tree. However, they have not included any methods based on large-scale text-to-molecule models, many of which are mentioned in the related works section. Therefore, it is unclear whether the improved performance of HI-Mol is attributable to the utilization of large-scale text-to-molecule models.

**Questions:**

See weakness

---

> ### Author Response · Authors · 2023-11-16
> **Response to Reviewer 89Qo**
>
> Dear Reviewer 89Qo,
>
> We sincerely appreciate your efforts in reviewing our manuscript. We respond to each comment in the following content. We carefully incorporated the discussions into the revised manuscript. We highlighted the revised contents in "$\text{\color{blue}blue}$" for your convenience to check.
>
> ---
>
> **[W1] Novelty. Multi-level molecule representation and embedding interpolation are well-established.**
>
>
> We politely disagree with the reviewer’s concern; although the high-level concepts of multi-level representation and embedding interpolation have been used for various purposes and domains in the literature, their usage for our own purpose could be novel and not straightforward. Specifically, we strongly believe that our novelty lies in (1) introducing the textual inversion for molecular generation and (2) advancing the existing textual inversion technique (which originally uses a single shared token in the image domain) by introducing multi-level tokens. Here, (2) is motivated by our new finding that a naïve adoption of textual inversion using a single shared token does not perform well in molecular generation. To alleviate this issue, we introduce a new concept, namely multi-level tokens, into the textual inversion framework to capture the complex distribution of molecules. Furthermore, we carefully design the sampling strategy based on the interpolation, since it is not straightforward to sample molecules from our newly proposed multi-level token embeddings. Therefore, our method is not a straightforward integration of previous techniques, but a novel framework for successfully adapting the textual inversion to molecular generation by introducing multi-level tokens into the textual inversion framework.
>
> ---
>
> **[W2] Rationale for adopting textual inversion appears somewhat nebulous. Graph representations are generally more adept at modeling molecular structures.**
>
> For clarification, we use SMILES strings as the representation of molecules because our method is built upon the state-of-the-art text-to-molecule model, MolT5 [1], that utilizes SMILES strings, not graphs. On the other hand, if there exist graph-based text-to-molecule models, our method is also applicable to them (then, we would use graphs as the representation of molecules). Namely, our method is agnostic to the underlying molecule representation of the foundation models, and we do not argue for the superiority of the molecule representation between SMILES strings and molecular graphs.
>
> ---
>
> **[W3] Absence of baselines based on large-scale text-to-molecule model.**
>
>
> Although our framework is built upon large-scale text-to-molecule models, they are not meaningful baselines. This is because it is not feasible to prompt all the training molecules to the mentioned large-scale text-to-molecule models due to the maximum token length of the input prompt. Specifically, while the maximum token length of the text-to-molecule model is only 1024 and 512 for [1] and [2], respectively, we have thousands of molecules and each SMILES representation of a molecule can sometimes consist of more than 100 tokens. Nevertheless, one can utilize the mentioned text-to-molecule models [1, 2] by providing the task description only (without providing the training molecules). For your information, the table below shows that our method indeed significantly outperforms such a naïve prompting of the text-to-molecule models on the HIV dataset. This confirms that our improved performance is not just due to the utilization of the large-scale text-to-molecule model, but to our carefully designed textual inversion framework. We added these respects of discussion and the experimental details in Appendix F of our revised manuscript.
>
> \begin{array}{l|cccccc}
> \hline
> \text{HIV} & \text{Active. $\uparrow$} & \text{FCD $\downarrow$} & \text{NSPDK $\downarrow$} & \text{Valid. $\uparrow$} & \text{Unique. $\uparrow$} & \text{Novelty $\uparrow$} \newline
> \hline
> \text{MolT5-Large-Caption2Smiles [1]} & 0.0 & 60.6 & 0.196 & \textbf{100} & 14.6 & \textbf{100} \newline
> \text{Text+Chem T5-augm-base [2]} & 0.0 & 62.2 & 0.188 & \textbf{100} & 90.2 & \textbf{100} \newline
> \hline
> \textbf{HI-Mol (Ours)} & \textbf{11.4} & \textbf{16.6} & \textbf{0.019} & \textbf{100} & \textbf{95.6} & \textbf{100} \newline
> \hline
> \end{array}
>
>
> ---
> [1] Edwards et al., Translation between Molecules and Natural Language, EMNLP 2022\
> [2] Christofidellis et al., Unifying Molecular and Textual Representations via Multi-task Language Modeling, ICML 2023

---

> ### Author Response · Authors · 2023-11-20
> **A Gentle Reminder**
>
> Dear Reviewer 89Qo,
>
> Thank you very much again for your time and efforts in reviewing our paper.
>
> We kindly remind that we have only three days or so in the discussion period.
>
> We just wonder whether there is any further concern and hope to have a chance to respond before the discussion phase ends.
>
> Many thanks, \
> Authors

---

> ### Comment · Reviewer_89Qo · 2023-11-22
>
> Dear Authors,
> I express my gratitude to the authors for their work. However, I still have some questions about certain aspects.
>
> 1. In the generation of the Hi-Mol method, are active molecules used as references? If so, have other generative models, including large text-to-molecule models, been fine-tuned based on active molecules?
>
> 2. I still think SMILES alone is not enough. Many graph-based generative models can also perform molecular optimization tasks. Examples of such models include Graph-GA, DST, RetMol, etc.

---

> > ### Author Response · Authors · 2023-11-23
> > **Response to Reviewer 89Qo**
> >
> > We sincerely appreciate your thoughtful response to our rebuttal. We respond to each additional comment in the following content.
> >
> > ---
> >
> > **In the generation of the Hi-Mol method, are active molecules used as references? If so, have other generative models, including large text-to-molecule models, been fine-tuned based on active molecules?**
> >
> > For clarification, our proposed hierarchical textual inversion framework and all other baseline generative models in Table 1 of our manuscript are trained using active molecules. On the other hand, for large text-to-molecule models in the initial rebuttal response, we prompt the task description into the model without such fine-tuning, due to the maximum token length of the input prompt. Here, we did not try fine-tuning large text-to-molecule models because fine-tuning such large-scale generative models with low-shot data is known to be difficult to perform well, e.g., see [1, 2, 3]. To address the issue, the textual inversion technique has successfully adopted low-shot data to large-scale generative models in the image domain. Inspired by this recent breakthrough, we carefully design a molecule-specific textual inversion framework for low-shot molecular generation.
> >
> >
> > Nevertheless, for your information, we fine-tuned large text-to-molecule models with active molecules as reported in the table below, where HI-Mol significantly outperforms them. We added these respects of discussion and the experimental details in Appendix F of our revised manuscript.
> >
> >
> > \begin{array}{l|cccccc}
> > \hline
> > \text{HIV} & \text{Active. $\uparrow$} & \text{FCD $\downarrow$} & \text{NSPDK $\downarrow$} & \text{Valid. $\uparrow$} & \text{Unique. $\uparrow$} & \text{Novelty $\uparrow$} \newline
> > \hline
> > \text{MolT5-Large-Caption2Smiles (fine-tuned)} & 6.0 & 23.3 & 0.024 & \textbf{100} & \textbf{96.0} & \textbf{100} \newline
> > \text{Text+Chem T5-augm-base (fine-tuned)} & 5.8 & 22.4 & 0.023 & \textbf{100} & 95.4 & \textbf{100} \newline
> > \hline
> > \textbf{HI-Mol (Ours)} & \textbf{11.4} & \textbf{16.6} & \textbf{0.019} & \textbf{100} & {95.6} & \textbf{100} \newline
> > \hline
> > \end{array}
> >
> > ---
> >
> > **I still think SMILES alone is not enough. Many graph-based generative models can also perform molecular optimization tasks. Examples of such models include Graph-GA, DST, RetMol, etc.**
> >
> > Molecular graphs are often more preferable at modeling molecular structures because they explicitly consider connectivity between atoms. However, in the recent literature on molecular generation, SMILES-based methods are known to outperform graph-based methods, e.g., as observed and discussed in [4]. This is because (a) there is “enough” information in a SMILES string to define a molecule (i.e., a SMILES string corresponds to a single molecule), and (b) the modeling dimension of SMILES strings (1D) is smaller compared to graphs (2D). In our experiments, we also observed that SMILES-based baselines (e.g., STGG) outperform graph-based methods.
> >
> >
> > For molecular optimization tasks (as reported in Table 4 of our manuscript), we kindly remind you that we follow the evaluation settings of the most competitive baseline, STGG, to show the wide applicability of HI-Mol. Here, our method still outperforms several graph-based methods (e.g., MHG-VAE, GraphAF, and GraphDF). Also, the mentioned Graph-GA is reported to perform worse than our baseline, PS-VAE. Nevertheless, we will include a comparison with the mentioned molecular optimization works in the final draft.
> >
> > ---
> >
> > [1] Mo et al., Freeze the Discriminator: a Simple Baseline for Fine-tuning GANs, arXiv 2020\
> > [2] Zhao et al., On Leveraging Pretrained GANs for Generation with Limited Data, ICML 2020\
> > [3] Moon et al., Fine-tuning Diffusion Models with Limited Data, NeurIPSW 2022\
> > [4] Vignac et al., DiGress: Discrete Denoising Diffusion for Graph Generation, ICLR 2023

---

> > > ### Comment · Reviewer_89Qo · 2023-12-03
> > >
> > > Since the authors have committed to include these comparisons in the final version, I will raise my score now.

---

### Author Response · Authors · 2023-11-16
**General Response**

Dear reviewers and AC,

We sincerely thank all the reviewers and AC for your enormous effort and time spent reviewing our manuscript.

We also appreciate all the positive comments from reviewers: e.g., importance of the target problem (89Qo, 5Up6), the method being novel (5Up6, 9Hsy), strong empirical results (all reviewers), and well-written manuscript (89Qo, 9Hsy). We believe that our paper proposes a simple yet effective data-efficient molecular generation framework by introducing a molecule-specific textual inversion framework which captures hierarchical features of the molecular data.

Thanks to the valuable comments on our manuscript. We have carefully revised and improved the manuscript in response to your concerns. The following list consists of newly added discussions and experimental results in our revised manuscript.

- Updated figures to better describe our method and motivation (Figure 1, 2)
- Added references of the recent molecular generation works (Section 2)
- Clarified description of the method details (Section 3)
- Updated table to show results of the QM9 experiments without Grammar (Table 16 $\rightarrow$ Table 3)
- Added discussion of the ablation on the number of clusters (Appendix E)
- Additional comparison with frozen text-to-molecule models (Appendix F)
- Rearrangement of paragraphs and figures (Overall manuscript)

We highlighted the revised contents in "$\text{\color{blue}blue}$" for your convenience to check. We believe that HI-Mol can be a useful addition to the ICLR and molecular generation community. Also, we hope that the above updates clarify the effectiveness of our method.

Thank you very much!

Best,\
Authors.

---

### Meta-Review · Area_Chair_jKVT · 2023-12-05

**Metareview:**

The submission introduces HI-Mol, a hierarchical textual inversion strategy for molecular generation, showcasing its effectiveness in improving data efficiency and performance in various molecular tasks. Reviewers generally commend the paper's articulation, systematic presentation, and empirical results demonstrating HI-Mol's superiority. They highlight the importance of the tackled problem and acknowledge the novelty of applying textual inversion to the molecular domain. Concerns center around the paper's clarity in describing the method, insufficient detail in figures, and the need for clearer comparisons against existing methods, especially graph-based ones. The absence of crucial baselines and certain comparisons has been noted. Some reviewers suggest clearer motivation in the introduction and rearranging figures/tables for better readability. Reviewer suggestions post-rebuttal mostly focus on additional comparisons with graph-based methods, inclusion of related works, and providing explanations for specific observations in the experimental results. While some concerns have been addressed in the rebuttal, others remain unaddressed, prompting reviewers to maintain their scores or offer slight score increases.

**Justification For Why Not Higher Score:**

Lack of detailed explanations in figures and tables, along with incomplete comparisons against related works, contributed to reservations among reviewers.

**Justification For Why Not Lower Score:**

N/A

---

### Decision · Program_Chairs · 2024-01-16

Reject